# Potential Role of Protein Kinase FAM20C on the Brain in Raine Syndrome, an In Silico Analysis

**DOI:** 10.3390/ijms24108904

**Published:** 2023-05-17

**Authors:** Icela Palma-Lara, Patricia García Alonso-Themann, Javier Pérez-Durán, Ricardo Godínez-Aguilar, José Bonilla-Delgado, Damián Gómez-Archila, Ana María Espinosa-García, Manuel Nolasco-Quiroga, Georgina Victoria-Acosta, Adolfo López-Ornelas, Juan Carlos Serrano-Bello, María Guadalupe Olguín-García, Carmen Palacios-Reyes

**Affiliations:** 1Laboratorio de Morfología Celular y Molecular, Escuela Superior de Medicina, Instituto Politécnico Nacional, Ciudad de México 11340, Mexico; 2Instituto Nacional de Perinatología Isidro Espinosa de los Reyes, Ciudad de México 11000, Mexico; 3División de Investigación, Hospital Juárez de México, Ciudad de México 11340, Mexico; 4Unidad de Investigación, Hospital Regional de Ixtapaluca, Ixtapaluca 56530, Mexico; 5Departamento de Biotecnología, Escuela de Ingeniería y Ciencias, Instituto Tecnológico de Monterrey, Toluca de Lerdo 50110, Mexico; 6Departamento de Oncología Quirúrgica, Hospital de Gineco-Obstetricia 3, Centro Médico Nacional “La Raza”, Ciudad de México 02990, Mexico; 7Laboratorio de Farmacología Clínica, Hospital General de México, Ciudad de México 06720, Mexico; 8Coordinación de Enseñanza e Investigación, Clínica Hospital Instituto de Seguridad y Servicios Sociales de los Trabajadores del Estado, Huauchinango 73177, Mexico; 9Departamento de Patología Clínica y Experimental, Hospital Infantil de México Federico Gómez, Ciudad de México 06720, Mexico; 10Centro Dermatológico “Dr. Ladislao de la Pascua”, Ciudad de México 06780, Mexico

**Keywords:** FAM20C, Raine syndrome, brain defects, in silico analysis, gene ontology, pathways

## Abstract

FAM20C (family with sequence similarity 20, member C) is a serine/threonine-specific protein kinase that is ubiquitously expressed and mainly associated with biomineralization and phosphatemia regulation. It is mostly known due to pathogenic variants causing its deficiency, which results in Raine syndrome (RNS), a sclerosing bone dysplasia with hypophosphatemia. The phenotype is recognized by the skeletal features, which are related to hypophosphorylation of different FAM20C bone-target proteins. However, FAM20C has many targets, including brain proteins and the cerebrospinal fluid phosphoproteome. Individuals with RNS can have developmental delay, intellectual disability, seizures, and structural brain defects, but little is known about FAM20C brain-target-protein dysregulation or about a potential pathogenesis associated with neurologic features. In order to identify the potential FAM20C actions on the brain, an in silico analysis was conducted. Structural and functional defects reported in RNS were described; FAM20C targets and interactors were identified, including their brain expression. Gene ontology of molecular processes, function, and components was completed for these targets, as well as for potential involved signaling pathways and diseases. The BioGRID and Human Protein Atlas databases, the Gorilla tool, and the PANTHER and DisGeNET databases were used. Results show that genes with high expression in the brain are involved in cholesterol and lipoprotein processes, plus axo-dendritic transport and the neuron part. These results could highlight some proteins involved in the neurologic pathogenesis of RNS.

## 1. Introduction

*FAM20C* (family with sequence similarity 20, member C) is mapped at 7p22.3 and belongs to the FAM20 family (“family with sequence similarity 20”), which consists also of *FAM20A* and *FAM20B* (mapped at 17q24.2 and 1q25.2, respectively). All of them encode protein kinases that act on diverse substrates [1,2,3], but FAM20C is the only one with activity towards casein. FAM20C was previously known as the Golgi casein kinase (GCK) since it phosphorylated casein-serine residues in lactating breast glands. It is classified as an atypical kinase due to its lack of sequence similarity with canonical kinases. Its structural span is 584 aa long, it contains a signal peptide (N-terminal 1–22 aa) in the immature form, and from residue 354 to 565 it corresponds to a kinase domain (KD) that spans 222 aa [4]. It is localized in the endoplasmic reticulum (ER) and Golgi and is relevant in the secretome pathway since its kinase domain targets the secretory pathway. It can phosphorylate serine residue at S-X-E/pS motifs of a wide number of different targets on the secretory pathway. Furthermore, its substrate numbers are higher since it can phosphorylate the S-x-Q-x-x-D-E-E motif [5,6].

FAM20C is well known for its expression and action in mineralized tissues, specifically in the family SCPP (the secretory calcium-binding phosphoprotein), and fibroblast growth factor 23 (FGF23), mainly associated with biomineralization and phosphatemia regulation [7]. Although its principal targets are proteins involved in bone mineralization, other proteins of the secreted phosphoproteome (>90%), including close to 100 substrates of serum and cerebrospinal fluid, are FAM20C targets [6,8,9,10]. The functional annotation of substrates indicates a wide variety of roles [9].

Raine syndrome is caused by biallelic FAM20C loss-of-function (LoF) pathogenic variants, resulting in a congenital phenotype due to prenatal effects, recognized as a sclerosing dysplasia due to bone- and tooth-mineralization defects, which can be considered the typical or classical RNS phenotype. These defects result from hypophosphorylation of FGF23 and SIBLING proteins (small integrin-binding ligand, N-linked glycoprotein) in bones and teeth. The typical RNS phenotype includes cranial and long-bone osteosclerosis, choanal atresia/stenosis, hypophosphatemia, and soft-tissue ectopic mineralization. Besides the classical phenotype, some RNS patients develop a more subtle or mild phenotype, featured by few bone abnormalities at an older age [11].

Other members of FAM20 family are also associated with pathologies. FAM20A, although considered a pseudokinase because of its lack of activity, forms a heterodimer with FAM20C in order to promote its activity and secondary enamel-matrix protein phosphorylation. Despite its lack of activity, LoF pathogenic variants are associated with enamel and renal defects, known as enamel–renal syndrome (OMIM #204 690), with autosomal recessive inheritance, also called amelogenesis imperfecta type 1G AI1G, enamel–renal–gingival syndrome, MacGibbon syndrome, Lubinsky–MacGibbon syndrome, amelogenesis imperfecta, and gingival fibromatosis syndrome, associated with amelogenesis imperfecta nephrocalcinosis, gingival fibromatosis, and impaired tooth eruption [10].

FAM20B is considered a glycan kinase since it phosphorylates to galactosyltransferase II in the xylose residue of the glycosaminoglycan–protein-linkage region [12] to increase its galactosyltransferase activity during glycosaminoglycan maturation [13,14]. Biallelic pathogenic LoF variants have been reported only in two siblings, who presented a lethal Desbuquois-like neonatal dysplasia, featured by severe growth failure, long-bone shortening, and thoracic hypoplasia.

In humans, all FAM20 members are expressed ubiquitously, including neural tissues. *FAM20A* expression is very similar among different human tissues, including the brain. However, considering that FAM20A is expressed in the normal brain, individuals with amelogenesis imperfecta and gingival hyperplasia syndrome with a deficiency of FAM20A have not been reported to present neural impairments. *FAM20B* is enriched in the brain [6], but its deficiency outcomes in neural defects are unknown since it is not described in reported cases [15]. FAM20C has a high expression in the liver and kidney and lower expression in bone and the brain. However, the level of expression in the latter tissues is very similar, which could indicate relevant functions in neural tissues. Furthermore, although RNS is mostly featured as skeletal impairment, affected individuals present a wide phenotype variability, including different neurologic alterations. Some affected individuals present structural and functional brain defects, including cognitive impairment and seizures.

SIBLING family members are the main FAM20C targets, which consist of osteopontin, bone sialoprotein, dentin-matrix protein 1, dentin sialophosphoprotein, and matrix extracellular phosphoglycoprotein, with actions in bone and dentin for proper mineralization [16]. Moreover, two-thirds of members of the secreted proteins or phosphoproteoma in serum, plasma, and cerebrospinal fluid, are also phosphorylated by FAM20C. These proteins are involved in processes and functions like lipid binding and homeostasis, insulin-like growth-factor-binding protein, neuropeptides, the cystatin family, protease inhibitors and metalloproteases, calcium binding and homeostasis, adhesion, extracellular matrix, the complement system, and blood clotting. Furthermore, 75% of the phosphoproteins in cerebrospinal fluid contain the S-x-E/pS motif (>100 proteins). In the last several years, a better understanding of phosphorylation effects on targets not belonging to SIBLING proteins have been studied, such as calsequestrin 2 (CSQ2), histidine-rich calcium-binding protein (HRC), matrix-interacting molecule 1 (STIM1), sortiline 1 (SORT1), and proprotein-convertase subtilisin 9 (PCSK9) [17]. It has been reported that FAM20C is also dysregulated in cancer, which leads to broader functions than those described for RNS [4].

The identification of FAM20C in cerebrospinal fluid and other tissues, the high number of targets, and the RNS neurologic phenotype support relevant FAM20C functions outside bone, including neural tissues. To date, it is unknown whether its deficiency results in hypophosphorylation of some brain FAM20C protein-targets–interactors and whether this could result in some molecular-pathway dysregulation. Identifying these data could highlight potential pathogenic mechanisms involved in RNS, but developmental and postnatal stages should be considered since structural and functional brain features are detected in some RNS cases. Therefore, the purpose of this work is to describe the neural defects reported in patients with RNS, to understand their frequency and type, and to evaluate the potential role of FAM20C in the brain, including its expression, its targets–interactors, and the possible pathways affected by its deficiency through an in silico analysis.

## 2. Results

### 2.1. Identification of Neurologic Alterations in RNS

#### 2.1.1. Identification of Structural and Functional Brain Defects in Reported RNS Cases

Seventy-two RNS cases were identified (January 2023). Neurologic alterations were described in 32 cases, including 11 lethal and 21 non-lethal cases, which represent a frequency close to 44%, with functional impairment in 22 cases and structural brain defects in 10 cases. Functional defects were more frequent than structural defects in non-lethal cases (21 vs. 4), and structural defects were more frequent in lethal cases (7 vs. 3). Functional defects included developmental delay (12 cases), followed by hypoacusia (11 cases) and seizures (6 cases). Seizures in RNS cases were related to low calcium levels in some and were of different types, including either focal or generalized, but a specific cause was not described or identified. Structural anomalies were located in the cortex (5 cases), corpus callosum, and cerebellum (4 cases each), as well as in the white matter (3 cases). Cortical defects were variable, with an abnormal gyral pattern, lissencephaly, dysplasia, and disorganization of cortical layers. Corpus-callosum defects included agenesis and hypoplasia. Table 1 lists all the functional and structural defects, considering both lethal and non-lethal patients, and includes the brain structures involved. Microcephaly is a frequent feature in RNS patients, but we did not consider it in structural brain defects. Among the non-lethal cases, only two presented structural defects, one associated with functional defects, whereas the other had no functional defect described (with non-visualization of the pituitary gland and mild prominence of the parietal and temporal horn of the left ventricle) [18].

#### 2.1.2. Identification of Reported Pathogenic Variants in RNS Individuals with Neurologic Alterations

FAM20C variants reported in each RNS case with functional and/or structural brain defect were identified. From 11 lethal cases, 8 had molecular diagnosis, whereas only 1 out of 21 non-lethal cases lacked variant identification. Identified variants were classified according to DNA and protein type and the exonic/intronic location, including those located within/outside of the kinase domain (KD). KDº was used to denote that the variant resulted in absence of the domain, KD* when the variant was located within the KD, and KD** when the variant was outside of KD. Of the eight lethal cases, three had two KDº alleles, two had two KD*, two had two KD**, and one had one allele KD* and one KDº. Non-lethal cases showed two KD* alleles in ten cases, six had two KD** alleles, and only one had two KDº alleles.

### 2.2. FAM20C Targets and Interactors

The identification of FAM20C targets and interactors was based on SIBLING proteins and FGF23, phosphoproteome analysis [9], and the BioGRID database [37]. Ninety-one targets were obtained from Tagliabracci’s report and 188 interactors from bioGRID (186 with physical evidence, one with genetic evidence, and one with more than one evidence type). Together with SIBLING proteins, 16 overlapped and were depurated (SPP2, APOA5, FGA, FGG, AHSG, APOB, DSPP, DMP1, IBSP, SERPINC1, IGFBP1, PCSK9, GPC3, KNG1, EVA1A, and SERPIND1), and 282 genes remained for subsequent analysis. Appendix A includes a list of all FAM20C targets–interactors identified.

### 2.3. Brain Expression of FAM20C Targets and Interactors

From 282 FAM20C targets and interactors, 263 showed expression in at least one brain structure (value > 1nTPM) according to the HPA database. Although most of them were expressed, the majority had low expression. The 10 top genes with higher expression or that were brain enriched were *TSC22D1*, *HSPD1*, *TIMP1*, *ATP1A2*, *APP*, *LGALS1*, *EEF1G*, *SPP1*, *APOE*, and *CLU* (with an nTPM value of 10,199 in the medulla oblongata, 10,098 in the hypothalamus, and 9359 in the white matter). The genes without expression (nTPM <= 1) were *AFP*, *BPIFB2*, *SPP2*, *APOA5*, *FGA*, *FGG*, *AHSG*, *APOB*, *DSPP*, *DMP1*, *SERPINC1*, *IGFBP1*, *AKAP8*, *GPC3*, *KNG1*, *EVA1A*, and *SERPIND1*, which had no neural functions. Figure 1 shows the expression level by brain structure.

As for the members of the FAM20 family, all were expressed in the brain. FAM20C was present in all structures, but was higher in the medulla oblongata (11.6 nTPMs), followed by the thalamus (10.1 nTPMs), midbrain (9.1 nTPMs), hypothalamus (7 nTPMs), white matter (6.9 nTPMs), amygdala (6.6 nTPMs), pons (6.5 nTPMs), cerebral cortex (6.1 nTPMs), spinal cord (6.1 nTPMs), basal ganglia (5.6 nTPMs), cerebellum (4.7 nTPMs), hippocampal formation (3.8 nTPMs), and olfactory bulb (3.1 nTPMs). FAM20A had a very low expression (nTPM < 4) in the thalamus, hypothalamus midbrain, and pituitary gland, and FAM20B had a low expression in all the brain structures (nTPMs between 12.7–18). FAM20C had higher expression than FAM20A and B (nTPMs as high as 18.3), being higher in the pituitary gland and basal ganglia. The expression of each target and interactor protein is shown in Figure 1a,b.

The level of FAM20C expression with regards to its targets–interactors was low, as it was in position 228 from high to low values. Since the cerebral cortex, cerebellum, and white matter are the structures with the most defects in RNS, we identified which genes had the highest expression in these structures. The corpus callosum was not considered because of its absence in HPA. The top 10% of genes with the highest expression are shown in Table 2. The enriched expression was shared for most of them in the cortex, cerebellum, and white matter, in addition to other structures. The most expressed genes found were *CLU*, *APOE*, *SPP1*, *EEF1G*, *LGALS1*, *APP*, *ATP1A2*, *HSPD1*, *TSC22D1*, *ATP1A1*, *EPHX1*, *TF*, *CALR*, and *HSP90B1*.

### 2.4. Brain-Cell-Type Expression Levels of Enriched Brain Genes

To identity potential relevant neuronal genes, those with the highest expression in the white matter, cortex, and cerebellum were analyzed in a public RNA-seq database of human-brain-cell subpopulations [38]. Differences between neurons, fetal astrocytes, mature astrocytes, microglia, oligodendrocytes, and endothelial cells were detected. FPKM (fragments per kilobase million) log2 values for nine enriched neuron genes including FAM20C in the different brain cells were plotted (Figure 2) and FAM20C expression was contrasted to each gene-expression value. *APP*, *HSP90B1*, *APLP2*, *ATP1A1*, *NCL*, *HSPD1*, *SET*, *TSC22D1*, *TCP1*, *SPP1*, *CALR*, *SORT*, *CLU*, and *ATP1A2* were the genes with the highest neuron expression.

### 2.5. Gene Ontology of FAM20C Targets–Interactors

To identify potential biological processes, functions, and components of FAM20C targets–interactors, gene-ontology (GO) analysis was performed. Genes with brain expression detected in HPA were associated with different terms.

GO processes identified many aspects related to cholesterol and lipoprotein events. This includes the terms "cholesterol efflux,” “lipoprotein metabolic process,” “transport,” “plasma-lipoprotein-particle assembly,” “cholesterol transport,” and “regulation of cholesterol esterification,” among others. In addition, the term “axo-dendritic transport” was identified (although in 31^st^ place), with the participation of APP (amyloid-beta precursor protein), AP3B1 (adaptor-related protein complex 3, beta 1 subunit), and AP3M1 (adaptor-related protein complex 3, mu 1 subunit). Table 3 shows the list of the most significant categories, and Figure 3 shows the categories according to their statistical significance. In each category and between them, APP, APOA5, APOE, and APOA2 had redundant functions, indicating a wide number of cellular functions.

GO functions with relevance in the brain include the term “cholesterol transporter activity,” with the participation of *APOE* and *APOA2*, and the term “protein kinase A regulatory-subunit binding,” with the participation of AKAP8 (a kinase (PRKA)-anchor protein 8) and ACBD3 (acyl-CoA-binding domain-containing 3). Table 4 and Figure 4 show the obtained list of gene-ontology functions and their statistical significance.

In the results involving component processes of brain aspects, the neuron-part category was identified. Genes in this category included *PPP3CA* (protein phosphatase 3, catalytic subunit, alpha isozyme), *CANX* (calnexin), *AP3B1* (AP-3 complex subunit beta-1), *AP3M1* (AP-3 complex subunit mu-1), *ATP1A1* (ATPase, Na^+^/K^+^ transporting, alpha 1 polypeptide), CDH2 (,Neural cadherin), *ATP1A2* (ATPase, Na^+^/K^+^ transporting, alpha 2 polypeptide), ASS1 (argininosuccinate synthase 1), *ADAM10* (Adam metallopeptidase domain 10), C4A (complement component 4a), and *COPA* (coatomer-protein complex, subunit alpha). Table 5 lists the gene-ontology-function categories and their statistical significance, which are shown in Figure 5.

GO terms in biological processes (Figure 3), molecular functions (Figure 4), and cellular components (Figure 5) were obtained through Gene Ontology enRIchement anaLysis and visuaLizAtion (Gorilla) based on 282 genes.The three GO terms converged in events related to cholesterol and lipoproteins. In addition, neural-specific categories such as the term “neuron part” in the GO cellular component and “axodendritic transport” in the GO biological processes were identified to have lower but significant *p*-values (FDR *p* < 8.58 × 10^−2^ and FDR *p* < 9.56 × 10^−3^ respectively).

### 2.6. FAM20C Target–Interactor Pathways

The pathway analysis for FAM20C targets–interactors was run with 263 genes through PANTHER, where 86 pathway categories were identified after filtering unclassified genes. The top 30 pathways are listed in Table 6, and Figure 6 shows the category pathways. The FAM20C protein targets and interactors identified are involved in different neural pathways, including neurodegenerative disorders such as Alzheimer’s and Parkinson’s diseases, and neurotransmitters such as enkephalin. The pathways involved also included genes that are relevant during development, such as FGF, EGF receptor, PDGF, VEGF, Wnt, and Notch signaling. Furthermore, genes involved in apoptosis pathways were identified.

### 2.7. Brain-Enriched FAM20C Targets–Interactors Associated with Disease

The most brain-enriched genes are associated with a wide number of diseases, according to the results from the DisGeNET database. The list of genes with the highest GDA score is shown in Table 7. The genes are associated with hundreds of different diseases. The gene with most disease hits was APOE, with 1049 traits or diseases, followed by APP with 486. All are associated with neurological diseases, some of them involving functional and/or structural brain defects.

## 3. Discussion

In the present review, 71 RNS cases were identified (January 2023). Neurologic alterations were described in 32 cases (~44%), including 11 lethal and 21 non-lethal cases, with functional impairments in 22 cases and structural brain defects in 8 cases.

### 3.1. Structural and Functional Neurological Defects in RNS

Functional defects were represented by developmental delay (12 cases), followed by hypoacusia (11 cases) and seizures (6 cases). There were structural anomalies in the cortex (5 cases), corpus callosum and cerebellum (4 cases each), and white matter (3 cases). Cortical defects were variable with abnormal gyral patterns, lissencephaly, dysplasia, and disorganization of cortical layers. Corpus-callosum defects included agenesis and hypoplasia. Table 1 lists all the functional and structural defects, including the brain structures involved. Since microcephaly can be related to different causes, it was not considered as a structural defect but was present in more than 50% of cases. In RNS, microcephaly could be related to different events such as hypoxia due to thoracic hypoplasia, craniosynostosis, or other causes. Considering all RNS cases, functional defects were more frequent with or without structural defects, but structural defects were in general related to functional impairments. It is relevant to note that one case with structural defects had no description of functional impairments. In non-lethal cases, developmental delay and seizures were the most frequents alterations. Summarizing these data, structural defects coexisted in lethal and non-lethal cases, with 33% and 13%, respectively.

#### 3.1.1. Structural Brain Defects

The frequency of structural defects has a general prevalence of 1–2 per thousand births [39]. Therefore, RNS patients have a higher risk of developmental brain impairments (44%), although it differs between lethal and non-lethal cases. For non-lethal cases, the risk is higher for functional defects, mainly developmental delay and seizures, whereas the risk for structural defects is higher for lethal RNS. The brain structural defects with higher risk involve the cortex, cerebellum, and corpus callosum. We estimate that the frequency of brain defects is probably higher than reported, especially in lethal cases, due to the lack of description of neurological aspects such as brain-image analysis and neurodevelopmental status, because of early mortality.

Cortex defects were the most frequently identified in RNS (4 cases), spanning different alterations such as abnormal gyral patterns, pachygyria, lissencephaly, disorganization of cortical layers, dysplasia, and atrophy. In neonates, cortical defects are associated with hypotonia, seizures, and early developmental delay and/or cognitive impairment. This is concordant with the described cases in RNS patients [40].

On the other hand, cortex defects result from developmental errors in early fetal life, since the time of their development starts around 18 days of gestation. Cortex development consists of three major events: cell proliferation, neuronal migration, and cortical organization. Further, cortex neuronal development spans neurogenesis, migration, post-migrational cortical organization, and circuit formation [41,42,43]. It depends on different cell-type participation, including radial glial cells, intermediate progenitor cells, and precursor cells located in the ventricular zone. Precursor cells migrate to populate the cerebral cortex mostly at 20–22 weeks of gestation; intermediate progenitor cells (IPCs) from the subventricular zone, generated from outer radial glial cells, contribute to both radial expansion and gyrification of brain. Radial glial cells act as a scaffold for proper migration and arrive at the cortical and subcortical levels to participate in the formation of superficial cortical layers [44,45]. Therefore, cortex-gyrus-formation defects can be associated with abnormal IPC and/or RGC actions and interactions of these cells, including migration defects and proliferation errors [40,46].

Human-brain development involves processes such as neurogenesis, synaptogenesis, and myelination, which are very specific spatially and temporally, resulting in different cellular types and specific cytoarchitecture [47]. These spatiotemporal events are concordant with differences in gene-expression patterns, although they have a wide transcriptional variability [48,49,50]. Neocortical areas in early to mid-fetal periods present a very high gene-expression variability [49,51].

The cortical developmental process has such a high complexity that different genetic and genomic approaches are applied and are necessary to gain a better understanding of normal and pathological aspects [52,53,54,55,56]. These approaches have led to identifying hundreds of coding and non-coding genes that participate in neurological disorders, including as the underlying mechanism in some cases. These data point to cortical defects in RNS beginning in early fetal development. However, analyzed FAM20C targets and pathways in this work are not associated with cortex development. Therefore, the underlying mechanism surely involves other targets and events.

The second most frequent defect in RNS was cerebellar hypoplasia. Cerebellar development starts very early, at 30 days post-conception [57]. It originates from the dorsal portion of the metencephalon and the neural folds and includes four steps: organization of the cerebellar territory, establishment of cerebellar progenitors (GABAergic and glutamatergic ones), migration of the granule cells, and formation of the cerebellar nuclei and circuitry [58]. The cerebellum neurons are also derived from the ventricular zone, in addition from the rhombic lip [59,60]. Congenital cerebellum defects are present frequently with other neural defects, such as cortical hypoplasia and corpus callosum agenesis, and sometimes are present as isolated defects. Syndromes with pontocerebellar hypoplasia allow candidate genes for cerebellum development to be identified, but little is known about the mechanism. Some involved genes include WNT/β-catenin signaling, *AHI1*, *FGF*, SHH pathway, FGF signaling, *ZIC1* and *ZIC4*, and BMP- and TGFβ-family genes, among many other candidates. On the other hand, genes relevant for cerebellum development include the fibroblast-growth-factor family (FGF8), WNT1, sonic hedgehog (SHH), bone-morphogenetic protein, and transforming growth factor-ß [60].

Developmental defects of the cerebellum can be present as part of more complex developmental syndromes in combination with other nervous-system defects such as cortical hypoplasia and corpus-callosum agenesis or, more rarely, as isolated defects. The cerebellum is mostly related to motor functions, including equilibrium, tone, posture, eye movements, and reflexes [61,62]. However, it is implicated in many other functions, such as cognition, social and behavioral conduct, visuo-spatial ability, visual and auditory sequential memory, visuomotor sequence, strategy planning, attention, and language. Cerebellum impairment results mostly in motor disturbances, including abnormal eye movements, dysmetria, tremor, ataxia, hypotonia, and vertigo, but is also related to deficits in cognitive, language, and social skills; executive functions; short-term memory; and visuo-spatial deficits [63].

In some RNS patients, besides cerebellum hypoplasia, they can present hypotonia and cognitive and language delay. Cerebellum hypoplasia has been related to clinical features such as hypotonia, which is associated with hyperreflexia and postural instability on standing [64]. As such, in RNS, these data could be associated with cerebellum hypoplasia or further functional deficits not identified to date, in addition to other structural or even functional brain defects.

In regard to the corpus callosum, defects were identified in only four RNS cases, which included hypoplasia and agenesis. Although few cases, the prevalence was higher than the 0.020–0.7% in the general population, but it is concordant with a prevalence of 1–3% when impaired neurodevelopment is present [65,66]. Its relevance relies on its function in connecting various regions of the contralateral hemisphere cortex, being the largest white-matter structure of the human brain. Callosal neurons originate from neocortex layers. Since it is formed between 13 and 17 weeks of gestation and total axon connections are completed at birth, its defects reflect an early fetal event. However, myelinization and redirection take place during the first years of life [67]. Defects can be partial or complete, including hypoplasia or agenesia, and can be present as a single defect or associated with others [68]. It can be part of different syndromic entities, including genetic, acquired, or chromosomal. Genetic causes are identified in 30–45% of cases (chromosomal causes in 10% and single gene mutations and copy-number variations in the remaining 20–35%). The involved genes have roles in axon guidance and post-guidance development, ciliary development, cell adhesion, proliferation, differentiation, and migration [67,69]. The number of genes and entities involved in corpus-callosum defects is abundant. Based on partial or complete defects, as well as the presence of other brain alterations, the clinical manifestations can develop from birth up to adolescence. Most affected individuals with isolated agenesis of the corpus callosum (ACC) are asymptomatic or present a wide variability of features, such as learning disability or cognitive impairment, muscle-tone alterations, alterations in the central osmoregulatory system, speech delay, attention disorders, cerebral palsy, or neurosensory impairments [70,71,72]. In RNS, individuals affected with CC dysgenesis were described in two lethal cases and two non-lethal cases that also presented cortical and functional defects. As such, functional defects can be related to cortical and callosal defects. In addition, FAM20C functions could be attributable to disruption of early brain development.

#### 3.1.2. Functional Brain Defects

Among functional defects in RNS, the most frequent include developmental delay, followed by hypoacusia and seizures. Based on the identified RNS cases, the risk for developmental delay in non-lethal cases is up to 52%, a risk that overcomes the general frequency reported. If developmental delay is present, the risk associated with structural brain defects is low, since the majority have no structural defects. In general terms, global developmental delay affects 1% to 3% of children, although its detection is limited in children younger than 5 years, since intellectual disability cannot be determined at this age. [73]. The causes are related to genetic disorders in nearly 47% and almost 28% to central-nervous-system malformations that have no identified etiology, but an underlying genetic cause cannot be ruled out [74]. As such, the frequency of developmental delay in RNS overcomes the risk related to an additional disorder. Intellectual-disability genes play roles in diverse basic cellular functions, such as DNA transcription and translation, protein degradation, mRNA splicing, chromatin remodeling, energy metabolism, and fatty-acid synthesis and turnover [75,76]. Single nucleotide variants (SNVs) and copy-number variants (CNVs) are associated with intellectual disability and brain malformations in >85% of cases, 50% are localized in known disease genes, whereas 50% are in unknown or candidate genes [76,77]. However, as the identification of genes and their functions in the brain increases, studies on this will continue. Furthermore, functional analysis of genes has shown phenotypic effects (for >90% of 5072 essential genes to HeLa cells), including multiple known and potential co-functional genes [78]. As such, new phenotypic effects for individual or co-functional FAM20C targets–interactors, including their pathogenic variants or structural variants, could be considered, and identified in the future [78].

Conductive sensorineural or mixed hearing loss is frequently reported in non-lethal RNS, with a frequency close to 45%, whereas in lethal cases it is reported in only one case. In comparison with the general congenital frequency of 2–3/1000, it is a prominent feature of RNS and points to a major role of FAM20C. The etiology of congenital hearing loss is genetic in up to 60% of cases, including 70% and 30% syndromic and non-syndromic entities, respectively. The number of syndromes with hearing loss is close to 500, and the related genes are close to 150 [79,80,81]. Functions of these genes are very heterogeneous, just like the associated clinical manifestations and phenotypes. Otosclerosis is the single most common cause of hearing impairment. Since RNS results in skeletal sclerosis with endochondral bone, bones in the middle ear may be affected, such as with otoesclerosis I (OMIM 166800), which features isolated endochondral-bone sclerosis of the labyrinthine capsule in adulthood, with effects on the stapediovestibular joint and stapes motion [82]. Furthermore, MEPE, a classic FAM20C target, is involved in otosclerosis [83], affecting the bone remodeling of the otic capsule. Based on these data, probable RNS audition is affected by bone and neural defects.

With respect to seizures, the risk for RNS patients is up to 10%. In children, seizures are among the most common neurologic disorders, considering that 1% of children experience afebrile seizures by 14 years of age and 5% by 6 years, and that epilepsy is the most common neurologic disorder in children, with a higher incidence during the first year of life [84]. Seizures can be associated or not with structural brain defects and are also related to acute brain injury or systemic insults, and a specific epileptic syndrome related to epilepsy is identified only in one third of children [85,86]. In RNS, seizures can be related to different events, such as hypoxia due to thoracic hypoplasia, structural brain defects, or calcium imbalances. This variability highlights some clinical approaches for RNS patients, without forgetting potential underlying molecular alterations. More complexity is added when considering a lack of epilepsy-syndrome identification in two thirds of cases and that more than 500 genes are related to seizures [87].

Regarding FAM20C variants in the reported RNS cases with functional and/or structural brain defects, there were differences in the possible activity effects between lethal and non-lethal cases. In lethal cases, the presence of both alleles showing complete LoF (amorphic variants) (KD°KD°) was the most frequent variant (3/8 cases), followed by two cases with one amorphic allele and one with a variant located in the KD (KD*KD°), and two cases with variants located in the KD. In non-lethal cases, the more frequent variants were represented by missense variants in both alleles (10 cases), which were likely hypomorphic since they were KD missense, followed by six cases with variants outside the KD, which were likely also hypomorphic. Only one non-lethal case with KD°KD° [34] was identified, although this patient presented a more severe phenotype than the other cases, considering facial, respiratory, and biochemical features. As such, there was a clear predominance of amorphic variants in lethal cases and hypomorphic alleles in non-lethal cases. However, it is more complex than this because some patients presented variants with different effects in both groups. Therefore, modifier genes should be considered, as previously suggested.

### 3.2. Brain Expression of FAM20C Targets–Interactors

The expression of FAM20C targets and interactors was higher than FAM20C, since 227 of them were above the expression level of FAM20C. Among the 10% of genes with the highest expression, most of them were highly expressed in the brain structures affected in RNS with more frequency, such as the cerebellum, cortex, and white matter. The similar expression pattern of these genes in these three structures may indicate a similar function, whereas the high expression may indicate their neurologic relevance. However, these genes have differential expression in different brain-cell types, and those enriched in neuron cells include *APP*, *HSP90B1*, *APLP2*, *ATP1A1*, *NCL*, *HSPD1*, *SET*, *TSC22D1*, *TCP1*, *SPP1*, *CALR*, *SORT*, *CLU*, and *ATP1A2*, all with different and relevant functions, as described below.

With respect to FAM20C targe–interactor gene ontology, the potential biological processes, functions, and components showed aspects related to cholesterol and lipoproteins in an overlapped way. In the gene ontology of biological process, only the term "axo-dendritic transport” was more specific of neural tissue, with the participation of the *APP*, *AP3B1*, and *AP3M1* genes.

APP is amyloid-beta-precursor protein, a cell-surface receptor and transmembrane-precursor protein cleaved by β- and γ-secretases to generate amyloid-beta peptides. The cleavage results in secreted peptides with different functions. These are related to different functions, such as transcriptional activation when bonded to acetyltransferase complex APBB1/TIP60 to form amyloid plaques, as well as roles in bacteriocidal and antifungal activities [88]. APP is mainly associated with neurodegenerative disease, including cerebroarterial amyloidosis (cerebral amyloid angiopathy, APP-related and Alzheimer’s disease, familial 1). However, it is also expressed during brain development, with probable roles in neuronal-stem and progenitor-cell proliferation, neuronal differentiation, migration, neurite growth, axonal growth and guidance, axonal transport, and synaptogenesis. It is also related to synaptic plasticity and neuroprotection [89].

On the other hand, a lack of APP in animal models can result in different alterations, including structural brain abnormalities such as agenesis of the corpus callosum or hippocampal commissure. In synapsis, it results in a reduction of dendritic length, impairment of function, plasticity, and long-term potentiation (LTP) [90]. The effect on the glia is also affected since it results in astrogliosis in the hippocampus and cortex. These data are very similar to only one case with APP LoF, in an individual who presented microcephaly, hypotonia, developmental delay, thinning of the corpus callosum, and seizures [91].

*AP3B1* is the protein adaptor-related protein complex 3 subunit beta 1, which is part of the heterotetrameric adaptor protein 3 (AP-3) complex and has roles in sorting transmembrane proteins, facilitating the budding of vesicles from the Golgi membrane to lysosomes and related organelles, and interacting with the scaffolding protein clathrin. Pathogenic variants result in Hermansky–Pudlak syndrome type 2 (OMIM 608233), resulting in defective lysosome-related organelles. The phenotype includes facial dysmorphism, oculocutaneous albinism, platelet and T-lymphocyte dysfunction and neutropenia, and neurological alterations such as developmental delay or mild intellectual disability.

*AP3M1* is adaptor-related protein complex 3 subunit Mu 1, which is another subunit of the adaptor protein 3 (AP-3) complex. To date, there are 10 Hermansky–Pudlak-syndrome types. Type 10 (OMIM 617050) is caused by pathogenic variants of AP3M1 and other genes. All Hermansky–Pudlak-syndrome types feature autosomal recessive inheritance, characterized by oculocutaneous albinism, a bleeding diathesis, and granulomatous colitis, neutropenia, or fatal pulmonary fibrosis in some cases. Type 10 has a neurophenotype that features developmental delay, different kinds of seizures, truncal hypotonia, dystonia, and structural brain alterations such as cerebral atrophy and delayed myelination [92]. RNS neurophenotypes share some features with AP3B1 and AP3M1 LoF, including development delay/intellectual disability, seizures, and hypotonia. AP3B1 and AP3M1 functions by sorting transmembrane proteins and the transport of vesicles from the Golgi membrane to lysosomes, similar to how FAM20C functions as a sorting receptor with roles in transport in the Golgi apparatus, lysosomes, and endosomes. Therefore, these genes involved in the term “axo-dendritic transport” could be converged in FAM20C functions such as sorting transmembrane proteins.

In the gene ontology of cellular components, the term “neuron part” is unique, as a specific term related to neuronal tissue. The genes involved in this include *PPP3CA*, *AP3B1*, *AP3M1*, *ATP1A1*, *CDH2*, *ATP1A2*, *ASS1*, *ADAM10*, *C4A*, *APOE*, *COPA*, *APP*, and *CLU*. Some of these genes are associated with neuronal alterations.

PPP3CA is the catalytic subunit A of a calcium-dependent protein phosphatase called calcineurin. Although it has ubiquitous expression, it is enriched in the brain and plays a role in the phosphorylation of synaptic receptors such as NMDARs and GABAARs.

Pathogenic variants result in two entities, depending on whether they are LoF or gain-of-function (GoF) variants. LoF results in developmental delay and epileptic encephalopathy 91 (OMIM 617711), an autosomal-dominant disorder featuring facial dymorphism and neurologic alterations. The latter can include hypotonia, epilepsy, severe delayed development/intellectual disability, neurodevelopmental regression, speech impairment, spasticity, cerebral atrophy, white-matter impairments, and behavioral abnormalities. GoF causes arthrogryposis, cleft palate, craniosynostosis, and impaired intellectual development [77,93,94].

ATP1A1 and 2 are ATPase Na^+^/K^+^-transporting subunit alpha 1 and 2, respectively. They belong to the family of P-type cation-transport ATPases and to the subfamily of Na^+^/K^+^ ATPases. The Na^+^/K^+^ ATPase is composed of two subunits (alpha and beta), located at the cell membrane, for the regulation of gradients of Na and K ions across the plasma membrane. Na and K gradients are basic for sodium-coupled transport and for the electrical excitability of nerves and muscle. They interact with voltage-gated calcium channels to modulate calcium-dependent action-potential generation and after hyperpolarization [95,96,97].

LoF pathogenic variants of ATP1A1 result in different entities, including Charcot–Marie–Tooth 2 DD (CMT2DD, OMIM 618036), hypomagnesemia accompanied by seizures and cognitive delay (HOMGSMR2, OMIM 618314), complex spastic paraplegia (CSP), borderline learning impairment/sleep disorders/poor emotional control, and severe developmental delay/focal seizures. Charcot–Marie–Tooth type 2 is related to axonal loss in the nerve fibers, reducing the neuromuscular-junction function and the action-potential amplitude in muscle. The effects of some variants produce a downregulation of mRNA and protein, producing defects in the differentiation of preneuronal cells into mature neuronal cells and the absence of neurites [88]. Pathogenic variants of ATP1A2 result in different disorders, such as familial hemiplegic migraine type 2 (FHM2) (602481) and migraine, familial basilar (OMIM 602481). These disorders have autosomal dominant inheritance and feature migraine, hemiparesis, hemiplegia, hemihypoasthesia, confusion and coma, dysphasia/aphasia, and episodic cerebellar signs. Some cases develop mental retardation or seizures. ATP1A2 is also associated with other autosomal dominant entities such as alternating hemiplegia of childhood 1 (AHC1, OMIM 104290), developmental and epileptic encephalopathy 98 (DEE98, OMIM 619605), and fetal akinesia, respiratory insufficiency, microcephaly, polymicrogyria, and dysmorphic facies (FARIMPD, OMIM 619602). These disorders share many neurological features, such as hemiplegia, development delay, cerebellar symptoms, and seizures. Only FARIMPD is related to structural brain defects in the cortex such as gyral abnormalities, enlarged ventricles, corpus-callosum and cerebellum hypoplasia, and further cortical and meningeal arterial calcifications [98]. ATP1A2 is highly expressed in the brain and is located in a perisynaptic distribution within cortical astrocytes, most likely associated with glutamatergic synapses. It mediates mechanisms between neuronal activity and energy metabolism. Its dysregulation results in altered calcium-dependent excitability in noradrenergic neurons [99], as well as reduced burst activity in neurons [100,101].

CDH2 is cadherin 2, a protein involved in calcium-dependent cell adhesion, with roles in development of the nervous system, cartilage, and bone, as well as left–right asymmetry establishment [102]. Pathogenic variants result in agenesis of the corpus callosum and cardiac, ocular, and genital syndrome (OMIM 618929), an entity featuring craniofacial dysmorphisms, global developmental delay/intellectual disability, corpus-callosum agenesis or hypoplasia, and ocular, cardiac, and genital anomalies. Neurological alterations include functional defects such as language delays, hypotonia/hypertonia, spasms, and seizures, among others, including behavioral/psychiatric features. Structural defects also include cerebellar-vermis hypoplasia, absent septum pellucidum, periventricular heterotopia, and megacisterna magna, among other defects [103,104].

In neurodevelopment, it has different roles in the neural tube, synaptogenesis, synapse plasticity, and glial-guided migration. It also has post-developmental functions in synaptic-vesicle endocytosis regulation and the proper generation, maintenance, and remodeling of synapses [102,105,106].

*ADAM10* is disintegrin and metalloproteinase domain-containing protein 10, a member of the ADAM family that corresponds to surface proteins with potential roles in cell adhesion and protein proteolysis. It is ubiquitous but has a high expression in the brain, including during neurodevelopment. It is expressed in different brain cells, including neurons, oligodendrocytes, and astrocytes. It probably participates in the cleavage of proteins from cell-adhesion molecules to membrane receptors such as cadherins, ephrins, amyloid-precursor protein (APP), prion proteins, and Notch receptors, as well as neuronal-adhesion molecules and the L1-adhesion molecule, for proper shedding and degradation. As such, ADAM10 is related to Alzheimer’s disease and neurodevelopment. ADAM10 plays an essential role during development. It is very relevant since its deficiency is lethal in mice during the fetal period [107], whereas its inactivation in neural-progenitor cells (NPCs) and NPC-derived neurons and glial cells results in defects in the cortex and ganglionic eminence and perinatal lethality [108]. Deficiency of ADAM10 in the postnatal brain produces increased mortality, associated with seizures, neuromotor and learning disabilities and related to impaired synaptic function, as well as astrogliosis, microglia activation, and alterations in the number and morphology of postsynaptic spines. These effects are mediated by a reduction of NMDA receptors subunit 2A and 2B and reduced-shedding cell-adhesion proteins such as Notch1, N-cadherin, nectin-1, and APP [108,109]. In addition, the functions of APP are related to ADAM10A as α-secretase, which cleavages APP to originate sAPPa, a neuroprotective APP-derived fragment, while preventing BACE-1 cleavage and lowering the formation of amyloid β (neurotoxic beta peptides) [110,111]. ADAM10 is associated with Alzheimer’s disease 18 (OMIM 615590), and reticulate acropigmentation of Kitamura (OMIM 615537).

CLU, APO A2, APOL1, and APOE were identified in different gene-ontology processes in relation to cholesterol homeostasis. Cholesterol is essential for the nervous system, since the brain is the richest organ in cholesterol and contains up to 25% of the total cholesterol of the body. As total of 80% of this cholesterol is in myelin sheaths and 20% corresponds to cell-plasma membranes [112]. Cholesterol also contributes to axon structure, dendrites, and synapses, and is a precursor of steroid hormones. Cholesterol is essential for neuronal development and functions, since its deficiency in neurons affects neurotransmission. As such, cholesterol-metabolism defects lead to structural and functional defects in the central nervous system, among other features. In adults, cholesterol deficiency is related to disorders such as Huntington’s disease, Alzheimer’s disease, and Parkinson’s disease [113].

Therefore, CLU, APOE, APOL1, and APOA2, identified in gene ontology with a role in cholesterol processes, could be relevant. APOE and CLU are expressed in the nervous system, whereas ApoA-II is synthetized in the liver and intestines and probably enters via the choroid plexus [114,115]. Although Apo A-II is considered to be synthesized in the liver, it was identified in HPA at very low levels in the cortex, hypothalamus, and white matter (1.1, 1.7, and 1.2 nTPM, respectively). To date, it is thought to be involved in HDL remodeling and cholesterol efflux [116], and its neural functions have not been described. However, low levels are related to Alzheimer’s disease, mild cognitive impairment, and neuroaxonal injury [116,117].

The *CLU* gene is clusterin, also known as apolipoprotein J (ApoJ). It plays different roles as a chaperone-like protein to clear misfolded proteins and protein aggregates from the cytosol and extracellular space, but is also involved in brain-cholesterol transport and metabolism, inflammation, apoptosis, protection of oxidative stress, and mitochondrial respiration, among others [118,119]. It is expressed ubiquitously, including in the central nervous system, where it is mainly expressed by astrocytes and secreted to the extracellular space. There are multiple CLU isoforms, which are located in different brain-cell types and compartments [120]. It is related to neurological diseases such as Alzheimer’s disease and Parkinson’s disease, schizophrenia, demyelination, accelerated cognitive decline, brain connectivity and structure in healthy individuals, repeated convulsions, and brain atrophy in mild cognitive impairment, among other pathologies [121,122,123]. It is involved in the clearance of amyloid and α-synuclein aggregates by astrocytes [124,125]; it participates in the regulation of neuronal proliferation, survival, and differentiation; and it probably has roles in synapsis [126,127,128]. The number of clusterin interactors is very high, including other apolipoproteins, complement factors, immunoglobulins, megalin, leptin, amyloid β-peptide, alpha-synuclein, prion protein, TGF beta-receptors, and stressed unfolded proteins [129].

Together with ApoE, clusterin is one of the main apolipoproteins in the brain parenchyma, but its role in lipid metabolism in the central nervous system (CNS) is not well understood. Although clusterin-lipidation status does not seem to affect amyloid binding, it may modify the affinity of clusterin for cell-surface receptors involved in uptake [129]. It is considered the third most prominent genetic risk factor for late-onset Alzheimer’s disease, just after apolipoprotein E (*APOE*) and bridging integrator 1 (*BIN1*) [122,130]. Furthermore, it is the second most abundant brain-expressed apolipoprotein, with a potential role in Alzheimer’s-disease pathogenesis [131]. However, its pro- or anti-amyloideogenic functions are not clear [129].

Higher levels of clusterin in the synapsis of APOE4 carriers contribute to synapse degeneration and synaptic accumulation of toxic amyloid beta in Alzheimer’s disease [127,132]. The Apo E4 variant correlates with low ApoE levels and with increased clusterin levels, probably as a compensatory mechanism to overcome low E4. A potential link between cholesterol and clusterin relies on a possible clusterin effect on the concentration of membrane cholesterol and the activity of many transmembrane receptors and enzymes [129].

*APOL1*, a member of the APOL family, is apolipoprotein L1, a unit of the high-density lipoprotein complex. It has roles in lipid metabolism and cholesterol transport [133]. It is expressed in the cytoplasm of both neuronal and glial cells. It is associated with neurological diseases such as frontotemporal dementia and schizophrenia. APOL1 is increased in FTD [134,135]. The former is also associated with cholesterol alterations [136]; however, the pathogenesis involved in neural disease is unknown.

*APOE* is apolipoprotein E, a major protein of the chylomicrons and essential for the normal catabolism of triglyceride-rich lipoprotein constituents. It has roles in cholesterol and phospholipid transport, as well as lipid transport between cells and tissues. Apolipoprotein E is synthesized mostly in the liver and intestines but is also highly expressed in the brain, mainly by astrocytes but also in neurons, oligodendrocytes, and microglia. Its function in the nervous system is the transference of phospholipids and cholesterol to promote axonal growth. It is also involved in neuroinflammation, tau hyperphosphorylation, Aβ aggregation, and clearance [137,138]. Apo E has three common genetic variants (Apo E2, E3, and E4). E4 is the major genetic risk for developing Alzheimer’s disease. Since the E4 allele results in a structural change that diminishes the ability to bind lipids, receptors, and amyloid-β, it favors an accumulation of cholesterol and impaired cholesterol and aggregate formation within the brain [139].

Besides cholesterol regulation by apolipoproteins, cholesterol-synthesis genes with LoF cause low levels of cholesterol in the brain and are associated with well-featured disorders that can have structural and functional neurological defects. These entities include Niemann–Pick type C, Smith–Lemli–Opitz, desmosterolosis, X-linked dominant chondrodysplasia punctata, CHILD syndrome, lathosterolosis, and hydrops-ectopic calcification–moth-eaten skeletal dysplasia. The pathology is related to a deficit of cholesterol and precursor actions during development, although the mechanism is unclear. All of these entities are associated with major developmental malformations, some of which include cortex, callosal, and cerebral defects, as well as intellectual disability. Some development pathways depend on cholesterol, such as the SHH pathway, since the addition of cholesterol is necessary for the activation of hedgehog-pathway proteins, which act on morphogen gradients in development [140].

Considering the functions of the proteins mentioned, it is concordant with the results of GenEDisease, which include the analysis of more enriched genes in the cortex, cerebellum, and white matter. There is concordance with the mechanisms involved in neurodegenerative disease and some entities with developmental delay, such as rasopathies. In addition, some of these proteins are involved in vesicle traffic and calcium homeostasis, although the latter is not mentioned in gene ontology.

## 4. Materials and Methods

### 4.1. Description of Neurologic Alterations in RNS Patients

To understand the frequency and type of neurologic alterations in RNS, data were collected from reported cases, identifying those with brain defects or neurocognitive-phenotype description, as well as cases with pathogenic variants, including the type and location in the gene and protein.

### 4.2. Identification of FAM2OC Targets–Interactors

The lists of FAM20C targets and interactors were obtained from known classic targets (SIBLING proteins and FGF23), from reported targets by Tagliabracci [9], and from protein interactors yielded by bioGRID (thebriogrid.org, accessed on 5 May and 5 November 2022). SIBLING proteins included were BSP1 or SPP1 (osteopontin), DMP1 (dentin-matrix protein 1), IBSP (integrin-binding sialoprotein), MEPE (matrix extracellular phosphoglycoprotein), DSPP (dentin sialophosphoprotein, including dentin sialoprotein (DSP), and dentin phosphoprotein (DPP). Additional targets such as FGF23, PCSK9, CASQ2, STIM1, SORT1, HRC, and other FAM20 family members were considered. BioGRID is a database of proteins and genetic and chemical interactions that includes interactors with physical or genetic (HTP) evidence, considering more than one evidence type. BioGRID Version 4.4.214 was used, and is updated to include interactions, chemical associations, and post-translational modifications (PTM) from 81,004 publications. BioGRID has a total number of 1,979,529 non-redundant interactions, 2,541,478 raw interactions, 13,057 non-redundant chemical associations, 29,417 raw chemical associations, 563,757 non-redundant PTM sites, and 57,396 unassigned PTMs.

### 4.3. FAM20 Members and Target–Interactor Expression Levels in the Brain

To understand FAM20C relevance in the brain, FAM20 family members and their targets and interactors were evaluated in the Human Protein Atlas database (HPA) (https://www.proteinatlas.org, accessed 30 November 2022). HPA provides gene expression and distribution of proteins in human tissues and cells. This tool uses data from normal and pathologic tissues for RNA-seq analysis and detection of proteins through immunohistochemical techniques. HPA contains RNA-seq of isolated human-brain-cell types, in addition to those of the mouse brain, including mRNA-expression data of human genes in 12 areas of the brain (amygdala, basal-ganglia cerebral cortex, hippocampal formation, thalamus, white matter, hypothalamus, cerebellum, pons, midbrain, medulla oblongata, and spinal cord). RNA-seq data were obtained for each target and interactor to evaluate their expression in different brain regions based on nTPM (normalized transcripts per million) values. The HPA classification of transcriptomics data is based on their tissue-specific, single-cell-type-specific, brain-region-specific, blood-cell-specific, or cell-line-specific expression into two different schemas: specificity category and distribution category. There is a total set of all nTPM values in 13 main regions of each mammalian brain, including human brains. HPA uses a cutoff value of 1 nTPM as a limit for detection across all tissues or cell types. RNA expression is classified as enriched when nTPM in a particular tissue/region/cell type is at least four times more than any other tissue/region/cell type, group enriched when nTPM in a group (of 2–5 tissues, brain regions, single cell types, or cell lines, or 2–10 blood-cell types) is at least four times more than any other tissue/region/cell line/blood-cell type/cell type), enhanced when nTPM in one or several tissues (1–5 tissues, brain regions or cell lines, or 1–10 immune-cell types or single cell types) is at least four times the mean of other tissue/region/cell types, low specificity when nTPM ≥ 1 in at least one tissue/region/cell type but not elevated in any tissue/region/cell type), and not detected when nTPM is <1 in all tissue/region/cell types [141]. The genes with the highest expression in three brain structures with more defects in RNS were identified and listed: cerebral cortex, white matter, and cerebellum.

### 4.4. Hierarchical Clustering

Brain-RNA expression of FAM20C targets and interactors was grouped in a heatmap, based on the log2 from nTPM data of the HPA database. Values corresponding to the “not detected” category were not considered for subsequent analysis (gene ontology and network analysis).

### 4.5. Brain-Cell-Type Expression Levels of Enriched Brain Genes

To ascribe the genes with the highest expression in the white matter, cortex, and cerebellum by cell type, the public RNA-seq database of human-brain-cell subpopulations was used (https://www.brainrnaseq.org/, accessed on 5 April 2023). FPKM log2 values for nine genes including FAM20C were identified as either neurons, microglia, fetal or mature astrocytes, oligodendrocytes, and endothelial cells. Each gene expression was contrasted with FAM20C expression [142].

### 4.6. Gene-Ontology Analysis

A gene-ontology (GO) analysis was performed using the GOrilla tool (*Gene Ontology enRIchment anaLysis and visuaLizAtion tool*, accessed on 23 November 2022) to determine overrepresented GO categories in the targes–interactors list including FAM20 members. A list of all genes analyzed in this study was used as a single ranked list of genes in GOrilla. After exporting the results of our gene dataset, the *p*-value was set to ≤0.05 [143].

### 4.7. Gene-Pathway Analysis

To identify common pathways in which FAM20C protein targets and interactors were involved, a network analysis of the previously detected brain-expressed genes was performed in terms of associated pathways using the online gene-ontology resource Protein Analysis THrough Evolutionary Relationships (PANTHER) (http://pantherdb.org/ accessed on 14 December 2022). PANTHER is a comprehensive knowledge base of evolutionary and functional relationships between protein-coding genes that utilizes tools to analyze large-scale genomic data, which allows the proteins under study to be classified considering evolutionary (protein class, protein family, subfamily) and functional groupings. Type-of-pathway categories and the gene-hit percentages were obtained [144].

### 4.8. Neurological Diseases Associated with FAM20C Targets–Interactors

To identify whether FAM20C targets–interactors with the highest expression in the brain are associated with neurological diseases or traits, the top 20 enriched genes were evaluated in DisGeNET database (https://www.disgenet.org accessed on 11 December 2022. Since brain defects in individuals with RNS most frequently involve the cortex, corpus callosum, cerebellum, and white matter, we identified which genes had the highest expression in each region, except for the corpus callosum, which is not included in HPA. The gene list is shown in Table 5. DisGeNET is an open-access platform collected from several sources and contains information on genes and variants associated with human diseases. The 7.0 version contains 1,134,942 gene–disease associations, including 21,671 genes and 30,170 diseases, as well as 369,554 variant–disease associations, including 194,515 variants and 14,155 diseases. Diseases also include disorders, traits, and clinical or abnormal human phenotypes. Data integrations of disease genomics and text-mined information are generated from expert-curated repositories, including ClinVar, ClinGen, Genomic England, Orphanet, and others, as well as GWAS catalogues, animal models, and scientific literature. The information is filtered and visualized based on several attributes and metric systems to prioritize genotype–phenotype relationships [145].

The gene–disease-association (GDA) score was used to prioritize the gene-association diseases for the input genes. It ranges from 0–1, with the highest value assigned if there is a high amount of evidence (level of curation, model organisms, number of publications supporting the association). We also considered the disease pleiotropy index (DPI), the evidence level (EL), and the evidence index (EI). The EL is adapted from a Genomics England PanelApp score corresponding to a score developed by ClinGen to evaluate the strength of evidence of a gene–disease association. The score ranges from strong (high) to moderate and low evidence (limited) of association. DPI considers whether multiple diseases associated with one gene or variant are similar or completely different diseases. The DPI ranges from 0 to 1 and depends on the number of associated disease classes (a lower value implies fewer diseases). The EI indicates the existence of contradictory results, with a score of 1 indicating that all the publications support the GDA and <1 indicating that there are contradictory results in associated publications.

## 5. Conclusions

We identified structural and functional defects in lethal and non-lethal RNS. The risk for functional defects is higher in non-lethal cases, represented by language/developmental delay, intellectual disability, and hypoacusia, whereas in lethal cases the risk is higher for structural cerebellum, cortex, and corpus-callosum defects. In lethal cases, the description of functional defects is very low, and alterations are probably more frequent but are not identified because of early lethality. More studies and a more detailed description of neurologic findings may change the frequency and/or provide more information on structural/functional brain defects. These may improve early detection and approaches to neurological risks.

The expression of the FAM20 family, as well as FAM20C targets and interactors, was identified in the brain in different structures. Some FAM20C interactors are brain and neuron enriched. Among the processes, functions, and molecular components identified in gene ontology, these participate in cholesterol metabolism and functions, vesicle traffic, neuron parts, and synapsis. As such, they probably have roles in neurodevelopment through cholesterol mechanisms and could impact differentiation and proliferation. Neurological aspects in RNS could be related to cholesterol and lipoprotein actions, which may be considered potential key pathogenesis components. Those targets–interactors with the highest brain and neuron expression may be a starting point to elucidate neural pathogenic mechanisms in RNS, yet, considering their present association with neurological diseases, could yield more information. Some of these proteins are also involved in calcium signaling or homeostasis. As FAM20C is involved mostly in phosphorylation of proteins for proper bone calcification, we consider that targets with roles in brain-calcium regulation may participate in neurological aspects in RNS, particularly those with the highest neuron expression.

Brain development requires complex molecular regulation, with the participation of a wide and coordinated number of proteins, including gene regulation, protein pathways, and protein–protein interaction networks, among others. Some of these events may be affected by lower phosphorylation in FAM20C deficiency, including development molecules. It would be interesting to understand FAM20C expression in different cell types and structures that are impaired in RNS, such as the cortex, corpus callosum, and cerebellum. This should be oriented towards processes or pathways in relation to brain development and neurocognitive functions involved in the RNS neurophenotype.

## Figures and Tables

**Figure 1 ijms-24-08904-f001:**
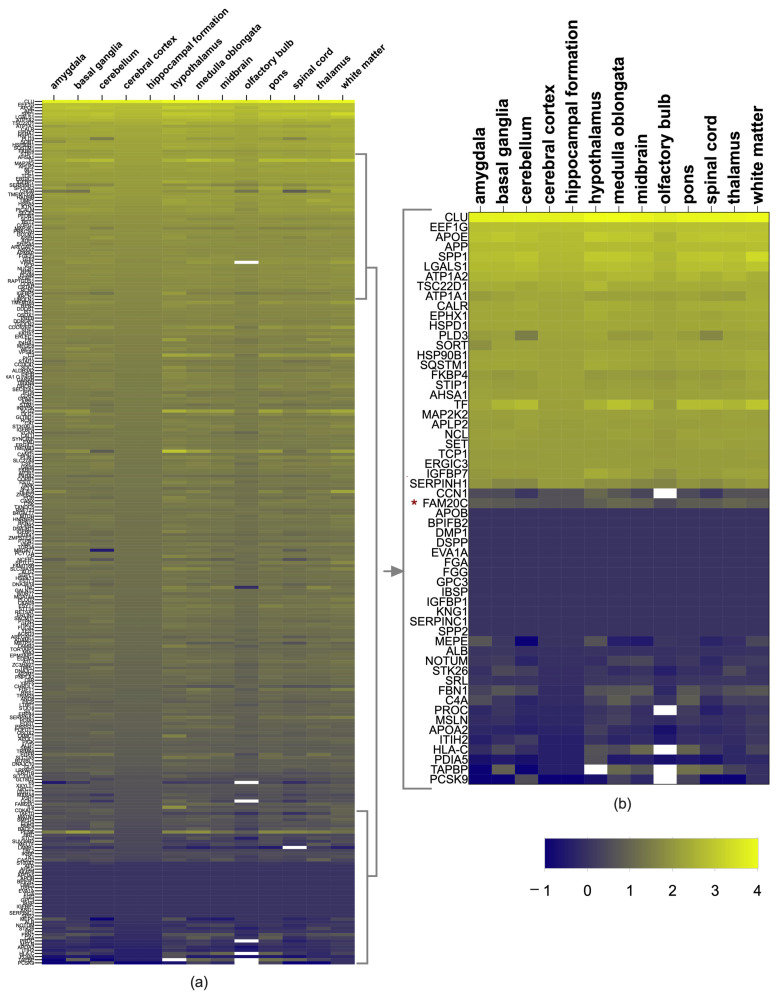
FAM20C target–interactor expression in brain regions. (**a**) Heatmap showing level of expression of FAM20C targets and interactors in different human-brain structures, using log2 values of nTPMs of each transcript from Human Protein Atlas. The upper genes have the highest expression in most regions (yellow color (top), whereas the blue corresponds to genes with lower expression (bottom). Cells in white correspond to values not reported. (**b**) Heatmap zoom of the 10% of genes with highest and lowest expression. Each row represents a gene and each column represents a human-brain region. Negative values correspond to low FPKM values.

**Figure 2 ijms-24-08904-f002:**
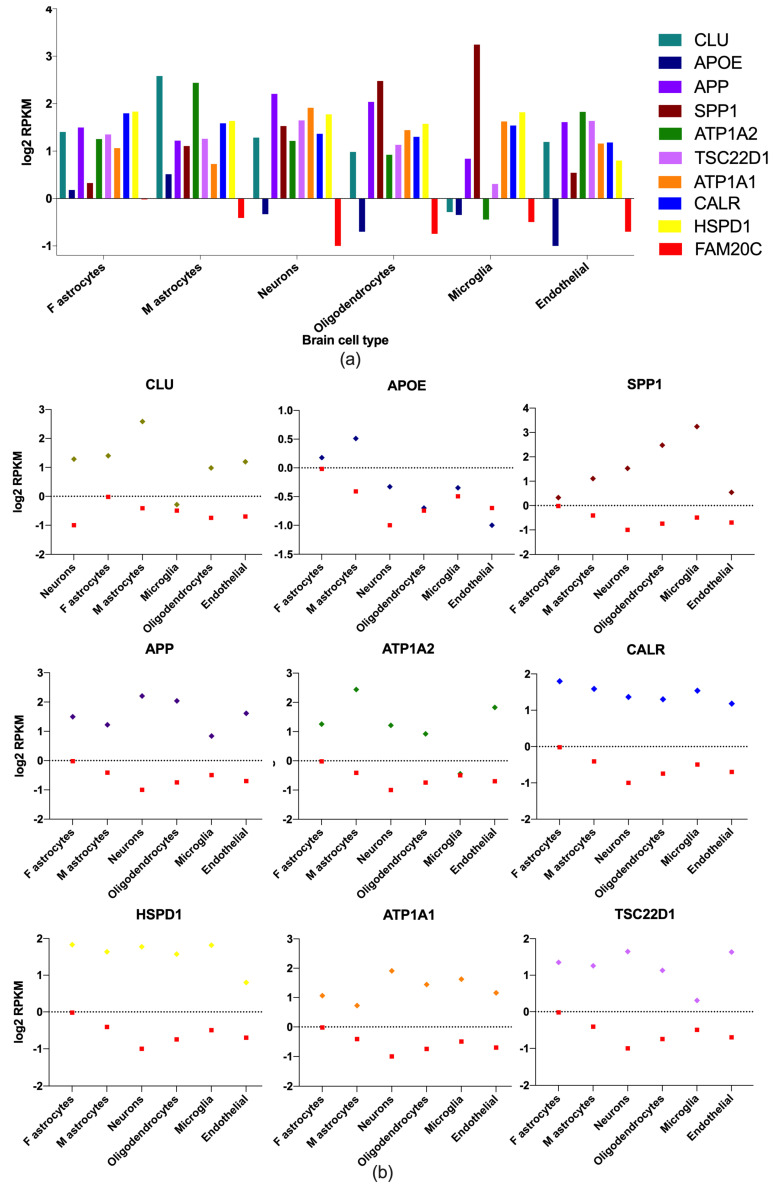
Brain-enriched FAM20C target–interactor expression in different brain-cell types. Gene-expression-level plots (RPKM log2) of FAM20C interactors, with the highest expression in neuron cells. (**a**) Plot showing nine of the highest-expressed genes in neurons and differences between brain-cell types. *APP*, *ATP1A1*, and *HSPD1* are the most neuron-enriched genes. (**b**) Expression-level plots (RPKM log2) showing each neuron-enriched gene contrasted to FAM20C expression. Data obtained from public human-brain RNA-seq dataset (www.brainrnaseq.org, accessed on 5 April 2023). Plotted negative values on Y−axis correspond to low RPKM values. F: fetal; M: mature.

**Figure 3 ijms-24-08904-f003:**
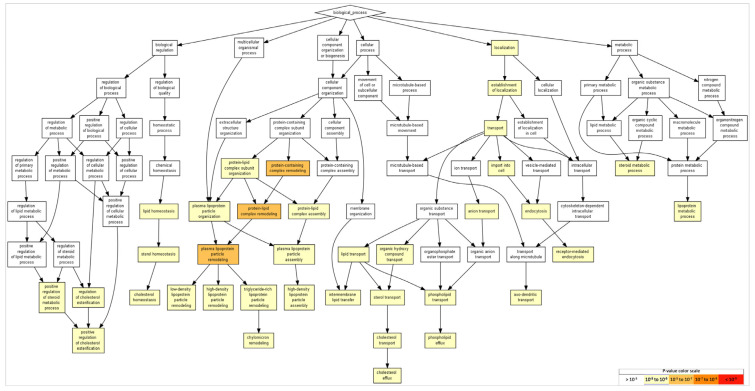
GO terms in biological processes. The GO terms reveal events related to cholesterol and lipoproteins. Terms with more significative *p*-values include protein-containing complex remodeling, protein–lipid-complex remodeling, and plasma-lipoprotein-particle remodeling (*p* = 10^−5^–10^−7^). As for neural events, the term “axodendritic transport” was detected (FDR *p* < 9.56 × 10^−3^). To identify enriched GO terms at the top of the ranked list of genes, *p*-values were obtained from the minimum hypergeometric score.

**Figure 4 ijms-24-08904-f004:**
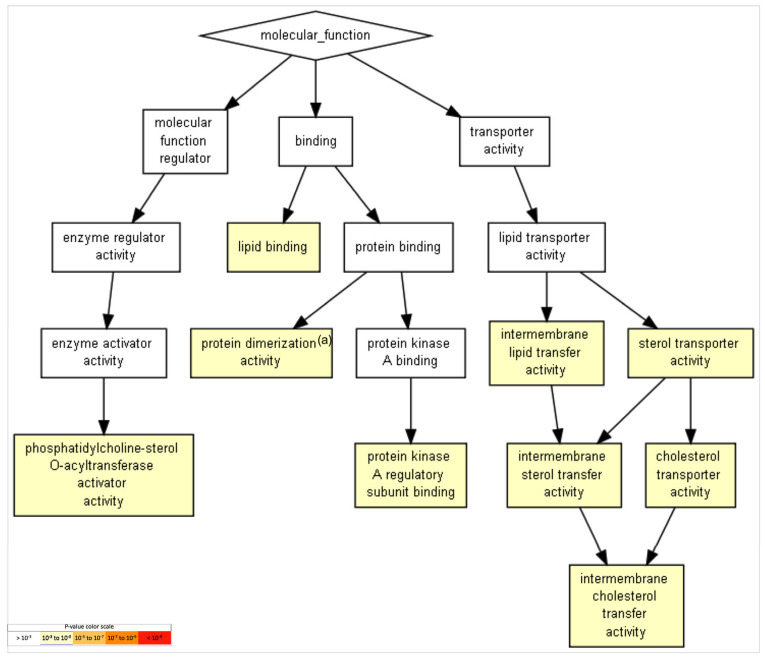
GO terms in molecular functions. GO terms with *p*-value < 10^−7^ for molecular functions were revealed and are presented. Terms with more significative *p*-values include “lipid binding,” “lipid-transfer activity,” and “sterol- and cholesterol-transport activity,” among others (*p* = 10^−3^–10^−5^). To identify enriched GO terms at the top of a ranked list of genes, *p*-values were obtained from the minimum hypergeometric score.

**Figure 5 ijms-24-08904-f005:**
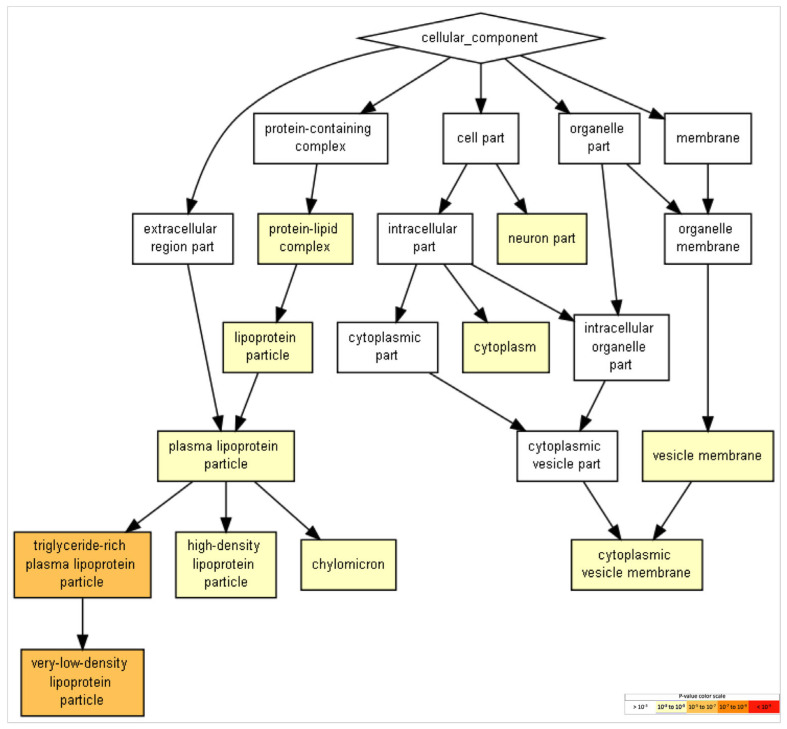
GO terms in cellular components. Terms with more significative *p*-values include “triglyceride-rich plasma-lipoprotein particle” and “very-low-density lipoprotein particle.” Events related to more neural-specific events include the term “neuron part” (FDR *p* < 8.58 × 10^−2^). To identifying enriched GO terms at the top of a ranked list of genes, *p*-values were obtained from the minimum hypergeometric score.

**Figure 6 ijms-24-08904-f006:**
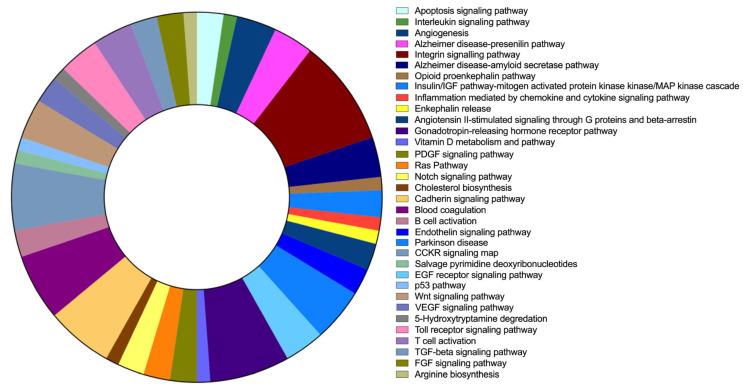
FAM20C target–interactor-pathway prediction. Results of PANTHER pathway-database analysis for FAM20C targets and interactors with brain expression. Each pathway is represented by a different color, whereas involved gene numbers in each pathway are reflected in the area size represented.

**Table 1 ijms-24-08904-t001:** Functional and structural brain defects and *FAM20C* variant types described in RNS patients.

Case	Lethal/Non-Lethal	Reference	Variant Description	Variant type	KD Effect	Variant Effect	Language Delay	Developmental delay/Learning difficulty	Seizures	Hypoacusia	Hydrocephaly	Visual impairment	Dystonic movements	Hypertonia	Hypotonia	Cortical defects	Corpus callosal Defects	Cerebellum Defects	Non-visualization of pituitary gland	Hypoplastic appearance of posterior part of the brain	White matter defects/Gliosis/astrogliosis	Encephalocele	Cortex	White matter/Glia	Cerebellum	Corpus callosum	Posterior brain	Pineal Gland	Functional only
1	L	[19]	ND	-	-	-												r			r			n	n				
2	L	[20]	ND	-	-	-																R							
3	L	[21]	ND	-	-	-										r		r					n		n				
4	L	[22]	46,XY.ar [hg19] 7p22.3 (36480- 523731)x0	GDel	KDºKDº	CG AM									X														A
5	L	[23]	c.1094G>A(Gly365Asp)/deletion	MS/GDel	KD*KDº	E6 HM/CG AM									X														A
6	L	[23]	c.1094G>A(Gly365Asp)/deletion	MS/GDel	KD*KDº	E6 HM/CG AM									x														a
7	L	[24]	c.456dupC(Gly153Argfs*56)/c.474delC (Ser159Profs*28)	NS/NS	KDºKDº	E1 AM/E1 AM										r	r						n			n			
8	L	[25]	c.1225C>T(p.Arg409Cys)	MS	KD*KD*	E6 HM										r		r			r		n	n	n				
9	L	[26]	c.1007T>G(p.Met336Arg)	MS	KD**KD**	E5 HM																r							
10	L	[27]	c.456delC(p.Gly153Alafs*34)	NS	KDºKDº	E1 HM												r							n				
11	L	[28]	c.1680C >A(p.Cys560Ter)	MS	KD*KD*	E10 HM		x		x		x																	a
		TOTAL			KDºKDº=3 KD*KD*=2KD**KD**= 2KD*KDº= 1		0	1	0	1	0	1	0	0	3	3	2	4	0	0	2	2	3	2	4	1			4
1	NL	[29]	c.1351G>A(p.Asp451Asn)	MS	KD*KD*	E7 HM	X	x	x	x	x																		A
2	NL	[29]	c.773T>A(p.IIe258Asn)/c.838 G>A(p.Gly280Arg)	MS	KD**KD**	E2 HM/E3 HM	X			x	x	x																	A
3	NL	[30]	c.803C>T (p.Thr268Met)/c.915C>A(p.Tyr305*)	MS/NS	KD**KDº	E3 HM/E4 AM	x																						A
4	NL	[30]	c.803C>T (p.Thr268Met)/c.915C>A(p.Tyr305*)	MS/NS	KD**KDº	E3 HM/E4 AM		x																					A
5	NL	[31]	c.1135G>A(p.Gly379Arg)	MS	KD*KD*	E6 HM			x		x		x	x	x														A
6	NL	[31]	c.784 + 5 G>C	Spe	KD**KD**	Int2 HM (sk E2)				x																			A
7	NL	[31]	c.784 + 5 G>C	Spe	KD**KD**	Int2 HM (sk E2)	x	x	x	x																			A
8	NL	[31]	c.784 + 5 G>C	Spe	KD**KD**	Int2 HM (sk E2)	x	x	x	x																			A
9	NL	[31]	c.1487C>T(p.P496L)	MS	KD*KD*	E9 HM						x																	A
10	NL	[32]	c.676T>A(p.Trp226Arg)	MS	KD**KD**	E2 HM	x		x	x																			A
11	NL	[32]	c.676T>A(p.Trp226Arg)	MS	KD**KD**	E2 HM	x			x																			A
12	NL	[18]	ND	-	-	-													r									nn	
13	NL	[33]	c.1219T>G(p.Tyr407Gly)	MS	KD*KD*	E6 HM				x																			a
14	NL	[2]	c.1228 T>A(p.Ser410Thr)	MS	KD*KD*	E6 HM	x	x	x						x	r	r			r			n	n		n			
15	NL	[34]	c.1351G>A(p.Asp451Asn)	MS	KD*KD*	E7 HM	x			x	x	x																	a
16	NL	[34]	c.1351G>A(p.Asp451Asn)	MS	KD*KD*	E7 HM	X																						A
17	NL	[34]	c.496G>T(p.E166X)	NS	KDºKDº	E1 AM	x																						A
18	NL	[35]	c.1645C>T (p.Arg549Trp)/c.863 + 5 G>C	MS/Spe	KD*KD**	E10 HM/Int3 HM		X			x	x				r	r						n			n			
19	NL	[28]	c.1680C>A(p.Cys560Ter)	MS	KD*KD*	E10 HM		X		x		x																	a
20	NL	[36]	c.1094G>T (Gly365Val)	MS	KD*KD*	E5 HM		X																					A
21	NL	[36]	c.1094G>T (Gly365Val)	MS	KD*KD*	E5 HM		x																					A
		TOTAL			KDºKDº=1 KD*KD*=10KD**KD**= 6KD**KDº= 2KD*KD**=1		11	11	6	10	5	5	1	1	2	2	2	0	1	1	0	0	2	1	0	2	0	1	18

L: lethal case; NL: non-lethal case. Cortical defects: atrophy, abnormal gyral pattern, dysplasia, disorganization of cortical layers. Corpus-callosum defects: agenesis, hypoplasia. Cerebellum defects: conical, hypoplasia. ND: non-detected. Gdel: genomic deletion. MS: missense. NS: nonsense. Spe: splicing effect. CG: complete gene. E1-10: exon 1-10. AM: amorphic HM: hypomorphic. Sk E2: skipping exon 2. KDº: kinase-domain absence. KD*: variant localized within kinase domain. KD**: variant located outside of kinase domain. Blue cells indicate presence of functional defects. Red cells indicate presence of structural defects. Black cells indicate affected regions. Yellow cells indicate the presence of only functional defects.

**Table 2 ijms-24-08904-t002:** Genes with enriched expression in the cortex, cerebellum, and white matter.

	Gene	Cerebral Cortex (nTPM Values)	Gene	White Matter (nTPM Values)	Gene	Cerebellum (nTPM Values)
1	CLU	8112.5	CLU	9359.1	CLU	7083.7
2	APOE	951	SPP1	1911	SPP1	752
3	SPP1	743	TF	812	APOE	680
4	EEF1G	727.9	LGALS1	792	EEF1G	623.2
5	LGALS1	701	APOE	763	LGALS1	566
6	APP	548	EEF1G	656.7	TF	536
7	ATP1A2	416.2	APP	605	APP	475
8	TIMP1	332	ATP1A2	297.4	ATP1A2	409.8
9	HSPD1	314.1	CALR	295.5	CALR	242.7
10	TSC22D1	298.2	HSP90B1	254.2	EPHX1	217
11	ATP1A1	278.3	ATP1A1	251.1	TSC22D1	214.5
12	AHSA1	274.8	EPHX1	247.2	HSPD1	209.2
13	FKBP4	269.2	HSPD1	243.6	ATP1A1	207.4
14	STIP1	263.1	TSC22D1	242.3	SQSTM1	190.4
15	EPHX1	251.2	SQSTM1	241.3	HSP90B1	189.8
16	SERPINH1	238.7	SCG2	230	APLP2	169
17	TF	235	CHGB	218	SPOCK2	156
18	PLD3	231.6	STIP1	204.1	MAP2K2	154
19	CALR	214.2	HLA-A	198.2	AHSA1	150.9
20	HSP90B1	205.9	PICALM	196.9	HADHB	145.4
21	SQSTM1	194.2	SPOCK2	196	CHGB	137
22	MAP2K2	192.2	FKBP4	187.2	FKBP4	133
23	SCG2	161	AHSA1	183.1	STIP1	127.7
24	APLP2	159	PLD3	182.7	PICALM	116.5
25	SPOCK2	140	HADHB	153.1	TIMP1	101
26	HADHB	139.4	APLP2	151	SERPINH1	52.1
27	PICALM	116.4	MAP2K2	146.5	PLD3	31.7
28	CHGB	103	TIMP1	141	SCG2	21.6
29	HLA-A	36.3	SERPINH1	114.4	HLA-A	20.1

**Table 3 ijms-24-08904-t003:** Gene-ontology process of FAM20C targets–interactors expressed in the brain.

No.	Description	*p*-Value	FDR *p*-Value	Involved Genes
1	Protein-containing-complex remodeling	1.97 × 10^−6^	8.49 × 10^−3^	*APOE*, *APOA2*, *ALB*
2	Protein–lipid-complex remodeling	1.97 × 10^−6^	4.24 × 10^−3^	*APOE*, *APOA2*, *ALB*
3	Plasma-lipoprotein-particle remodeling	1.97 × 10^−6^	2.83 × 10^−3^	*APOE*, *APOA2*, *ALB*
4	Protein–lipid-complex subunit organization	2.30 × 10^−5^	2.48 × 10^−2^	*APOE*, *APOA2*, *ALB*
5	Plasma-lipoprotein-particle organization	2.30 × 10^−5^	1.98 × 10^−2^	*APOE*, *APOA2*, *ALB*
6	Triglyceride-rich lipoprotein-particle remodeling	2.80 × 10^−5^	2.01 × 10^−2^	*APOE*, *APOA2*
7	Cholesterol efflux	2.80 × 10^−5^	1.72 × 10^−2^	*APOE*, *APOA2*
8	Lipoprotein metabolic process	1.08 × 10^−4^	5.81 × 10^−2^	*APOE*, *APOA2*, *APOL1*
9	Transport	1.87 × 10^−4^	8.95 × 10^−2^	*APOA2*, *COPG1*, *ARFGAP2*, *CANX*, *AP3M1*, *APLP2*, *ATP1A1*, *ATP1A2*, *CASQ2*, *ARFGAP1*, *CP*, *COPB2*, *ADAM10*, *BST2*, *ANO8*, *AUP1*, *AFP*, *CLU*, *CKAP4*, *PPP3CA*, *ARCN1*, *CAT*, *AP3B1*, *ASHG*, *APOL1*, *ASPH*, *CAND1*, *CALR*, *C3*, *APOE*, *COPB1*, *AKAP8*, *COPA*, *APP*, *ALB*
10	Plasma-lipoprotein-particle assembly	2.22 × 10^−4^	9.57 × 10^−2^	*APOE*, *APOA2*
11	Cholesterol transport	2.22 × 10^−4^	8.70 × 10^−2^	*APOE*, *APOA2*
12	Organic-hydroxy-compound transport	2.22 × 10^−4^	7.98 × 10^−2^	*APOE*, *APOA2*
13	Sterol transport	2.22 × 10^−4^	7.36 × 10^−2^	*APOE*, *APOA2*
14	Sterol homeostasis	2.22 × 10^−4^	6.84 × 10^−2^	*APOE*, *APOA2*
15	Protein–lipid-complex assembly	2.22 × 10^−4^	6.38 × 10^−2^	*APOE*, *APOA2*
16	Cholesterol homeostasis	2.22 × 10^−4^	5.98 × 10^−2^	*APOE*, *APOA2*
17	Establishment of localization	3.22 × 10^−4^	8.18 × 10^−2^	*APOA2*, *COPG1*, *ARFGAP2*, *CANX*, *AP3M1*, *APLP2*, *ATP1A1*, *ATP1A2*, *CASQ2*, *ARFGAP1*, *COPB2*, *CP*, *ADAM10*, *BST2*, *ANO8*, *AUP1*, *CLU*, *CKAP4*, *PP3CA*, *ARCN1*, *CAT*, *AP3B1*, *AHSG*, *APOL1*, *ASPH*, *CAND1*, *CALR1*, *C3*, *APOE*, *AKAP8*, *COPB1*, *COPA*, *APP*, *ALB*
18	Lipid transport	3.42 × 10^−4^	8.21 × 10^−2^	*APOE*, *APOA2*, *APOL1*
19	Import into cell	3.73 × 10^−4^	8.47 × 10^−2^	*CALR*, *APOE*, *APP*, *ALB*, *CANX*, *AP3M1*, *ATP1A1*, *ATP1A2*, *APOL1*, *ASHG*
20	Positive regulation of cholesterol esterification	5.11 × 10^−4^	1.10 × 10^−1^	*APOE*, *APOA2*,
21	Regulation of cholesterol esterification	5.11 × 10^−4^	1.05 × 10^−1^	*APOE*, *APOA2*,
31	Axo-dendritic transport	6.16 × 10^−4^	8.58 × 10^−2^	*APP*, *AP3B1*, *AP3M1*

Statistical significance for each category is shown, including *p*-value and FDR q-value.

**Table 4 ijms-24-08904-t004:** Gene-ontology functions of FAM20C targets–interactors expressed in the brain.

N	Description	*p*-Value	FDR q-Value
1	Lipid binding	4.33 × 10^−5^	3.52 × 10^−2^
2	Protein-dimerization activity	4.56 × 10^−5^	1.85 × 10^−2^
3	Intermembrane cholesterol-transfer activity	5.11 × 10^−4^	1.38 × 10^−1^
4	Intermembrane sterol-transfer activity	5.11 × 10^−4^	1.04 × 10^−1^
5	Intermembrane lipid-transfer activity	5.11 × 10^−4^	8.30 × 10^−2^
6	Sterol-transporter activity	5.11 × 10^−4^	6.92 × 10^−2^
7	Phosphatidylcholine-sterol O-acyltransferase-activator activity	5.11 × 10^−4^	5.93 × 10^−2^
8	Cholesterol-transporter activity	5.11 × 10^−4^	5.19 × 10^−2^
9	Protein kinase A regulatory-subunit binding	8.02 × 10^−4^	7.24 × 10^−2^

Categories with more statistical significance are shown from top to bottom according to *p*-value and FDR q-value.

**Table 5 ijms-24-08904-t005:** Gene-ontology components of FAM20C targets–interactors expressed in the brain.

N	Description	*p*-Value	FDR *q*-Value
1	Very-low-density lipoprotein particle	2.68 × 10^−6^	1.47 × 10^−3^
2	Triglyceride-rich plasma-lipoprotein particle	2.68 × 10^−6^	7.35 × 10^−4^
3	Lipoprotein particle	2.73 × 10^−5^	4.99 × 10^−3^
4	Plasma-lipoprotein particle	2.73 × 10^−5^	3.74 × 10^−3^
5	High-density lipoprotein particle	2.73 × 10^−5^	2.99 × 10^−3^
6	Protein–lipid complex	2.73 × 10^−5^	2.49 × 10^−3^
7	Chylomicron	2.80 × 10^−5^	2.19 × 10^−3^
8	Neuron part	1.40 × 10^−4^	9.56 × 10^−3^
9	Vesicle membrane	8.45 × 10^−4^	5.14 × 10^−2^
10	Cytoplasmic-vesicle membrane	8.45 × 10^−4^	4.63 × 10^−2^
11	Cytoplasm	9.77 × 10^−4^	4.87 × 10^−2^

Statistical significance for each category is shown, including *p*-value and FDR q-value.

**Table 6 ijms-24-08904-t006:** Potential FAM20C target–interactor pathways identified in PANTHER.

N	Pathway Category (Panther ID)	Number of Genes	% of Involved Genes	% of Involved Pathways
1	Apoptosis-signaling pathway (P00006)	2	0.70%	2.30%
2	Interleukin-signaling pathway (P00036)	1	0.40%	1.20%
3	Angiogenesis (P00005)	3	1.10%	3.50%
4	Alzheimer’s-disease presenilin pathway (P00004)	3	1.10%	3.50%
5	Integrin-signalling pathway (P00034)	8	3.00%	9.30%
6	Alzheimer’s-disease amyloid-secretase pathway (P00003)	3	1.10%	3.50%
7	Opioid-proenkephalin pathway (P05915)	1	0.40%	1.20%
8	Insulin/IGF pathway mitogen-activated protein kinase Kinase/MAP-kinase cascade (P00032)	2	0.70%	2.30%
9	Inflammation mediated by chemokine- and cytokine-signaling pathway (P00031)	1	0.40%	1.20%
10	Enkephalin release (P05913)	1	0.40%	1.20%
11	Angiotensin II-stimulated signaling through G proteins and beta-arrestin (P05911)	2	0.70%	2.30%
12	Endothelin-signaling pathway (P00019)	2	0.70%	2.30%
13	Parkinson’s disease (P00049)	4	1.50%	4.70%
14	EGF-receptor-signaling pathway (P00018)	3	1.10%	3.50%
15	Gonadotropin-releasing hormone-receptor pathway (P06664)	6	2.20%	7.00%
16	Vitamin D metabolism and pathway (P04396)	1	0.40%	1.20%
17	PDGF-signaling pathway (P00047)	2	0.70%	2.30%
18	Ras pathway (P04393)	2	0.70%	2.30%
19	Notch-signaling pathway (P00045)	2	0.70%	2.30%
20	Cholesterol biosynthesis (P00014)	1	0.40%	1.20%
21	Cadherin-signaling pathway (P00012)	5	1.90%	5.80%
22	Blood coagulation (P00011)	5	1.90%	5.80%
23	B-cell activation (P00010)	2	0.70%	2.30%
24	CCKR-signaling map (P06959)	5	1.90%	5.80%
25	Salvage pyrimidine deoxyribonucleotides (P02774)	1	0.40%	1.20%
26	p53 pathway (P00059)	1	0.40%	1.20%
27	Wnt-signaling pathway (P00057)	3	1.10%	3.50%
28	VEGF-signaling pathway (P00056)	2	0.70%	2.30%
29	5-Hydroxytryptamine degredation (P04372)	1	0.40%	1.20%
30	Toll-receptor-signaling pathway (P00054)	3	1.10%	3.50%
31	T-cell activation (P00053)	3	1.10%	3.50%
32	TGF-beta-signaling pathway (P00052)	2	0.70%	2.30%
33	FGF-signaling pathway (P00021)	2	0.70%	2.30%
34	Arginine biosynthesis (P02728)	1	0.40%	1.20%

**Table 7 ijms-24-08904-t007:** Diseases associated with FAM20C targets and interactors enriched in the cerebral cortex, white matter, and cerebellum.

Gene	Disease	Disease Class	Number of Associated Genes	GDA Score	EL Score	EI Score
HSPD1	Spastic paraplegia 13, autosomal dominant	Congenital, hereditary, and neonatal diseases and abnormalities; nervous-system diseases	1	0.92		1
SERPINH1	Osteogenesis imperfecta Type X		1	0.91	strong	1
TF	Congenital atransferrinemia	Congenital, hereditary, and neonatal diseases and abnormalities; nutritional and metabolic diseases	3	0.91	strong	1
APP	Alzheimer’s disease	Nervous-system diseases; mental disorders	3397	0.9		0.981
APOE	Hyperlipoproteinemia type III	Congenital, hereditary, and neonatal diseases and abnormalities; nutritional and metabolic diseases	28	0.8	strong	0.977
APOE	Alzheimer’s disease 2	Nervous-system diseases; mental disorders	16	0.8	strong	0.96
APOE	Lipoprotein glomerulopathy	Female urogenital diseases and pregnancy complications; male urogenital diseases	5	0.8	strong	0.95
ATP1A2	Migraine, familial hemiplegic 2	Nervous-system diseases	6	0.8		1
CALR	Primary myelofibrosis	Hemic and lymphatic diseases	282	0.8	limited	0.929
MAP2K2	Cardio-faciocutaneous syndrome	Congenital, hereditary, and neonatal diseases and abnormalities; skin and connective-tissue diseases; cardiovascular diseases	28	0.8	definitive	0.966
HADHB	Trifunctional-protein deficiency with myopathy and neuropathy	Congenital, hereditary, and neonatal diseases and abnormalities; nutritional and metabolic diseases; musculoskeletal diseases; nervous-system diseases; cardiovascular diseases	23	0.78	definitive	1
HSPD1	Leukodystrophy, hypomyelinating 4	Congenital, hereditary, and neonatal diseases and abnormalities; nutritional and metabolic diseases; nervous-system diseases	3	0.73		1
APOE	Alzheimer’s disease	Nervous-system diseases; mental disorders	3397	0.7		0.946
APOE	Coronary heart disease	Cardiovascular diseases	1576	0.7		0.966
APOE	Hypercholesterolemia	Nutritional and metabolic diseases	489	0.7		0.957
APOE	Hypertensive disease	Cardiovascular diseases	2322	0.7		0.897
APOE	Sea-blue histiocyte syndrome	Congenital, hereditary, and neonatal diseases and abnormalities; nutritional and metabolic diseases; nervous-system diseases; hemic and lymphatic diseases	5	0.7	strong	1
APP	Dementia	Nervous-system diseases; mental disorders	816	0.7	strong	0.964
APP	Cerebral amyloid angiopathy, app-related	Congenital, hereditary, and neonatal diseases and abnormalities; nutritional and metabolic diseases; nervous-system diseases; cardiovascular diseases	1	0.7		1
ATP1A1	Charcot–Marie–Tooth disease, axonal, type 2dd		2	0.7	strong	1
SQSTM1	Amyotrophic lateral sclerosis	Nutritional and metabolic diseases; nervous-system diseases	1114	0.7		0.974
MAP2K2	Melanoma	Neoplasms	3087	0.67		1
MAP2K2	Noonan syndrome	Congenital, hereditary, and neonatal diseases and abnormalities; skin and connective-tissue diseases; musculoskeletal diseases; cardiovascular diseases	85	0.64	limited	1
ATP1A2	Alternating hemiplegia of childhood 1		2	0.61		1
SQSTM1	Nonaka myopathy	Congenital, hereditary, and neonatal diseases and abnormalities; musculoskeletal diseases; nervous-system diseases	132	0.61	strong	1
APOE	Hyperlipidemia	Nutritional and metabolic diseases	472	0.6		0.957
APOE	Schizophrenia	Mental disorders	2872	0.6		0.78
APOE	Type IIa hyperlipoproteinemia	Congenital, hereditary, and neonatal diseases and abnormalities; nutritional and metabolic diseases	201	0.6		0.889
APP	Impaired cognition	Mental disorders	1630	0.6		0.98
APP	Cerebral hemorrhage with amyloidosis, hereditary, Dutch type	Congenital, hereditary, and neonatal diseases and abnormalities; nutritional and metabolic diseases; nervous-system diseases; cardiovascular diseases	20	0.6		1
CALR	Myelofibrosis	neoplasms; hemic and lymphatic diseases	163	0.6	limited	0.933

GDA: gene–disease associations. GDA score: score of gene–disease association. EL score: evidence level of GDA. EI score: evidence index of GDA.

## Data Availability

No new data were created or analyzed in this study. Data sharing is not applicable to this article.

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
