# Peer review of "Potential Role of Protein Kinase FAM20C on the Brain in Raine Syndrome, an In Silico Analysis"

_ijms, 2023, doi:10.3390/ijms24108904_

Round 1
Reviewer 1 Report
This manuscript attempts to correlate the pathophysiology of Raine Syndrome with sub-lethal mutations in Fam20C, expression of FAM20C targets and interactors in different human brain structures, and their roles in biological processes. As stated by the authors, the precise molecular defects downstream of biallelic Fam20C mutations in Raine Syndrome is not well understood but hypothesize that loss of Fam20C-kinase activity is expected to result in the hypo-phosphorylation of specific targets, alteration of the brain and/or CSF phosphoproteome(s), and thereby contribute to the developmental pathophysiology of this syndrome.
Overall Summary:
While the overall hypothesis is clearly stated and well justified, the analyses presented in the manuscript falls short of providing a significant mechanistic link between Fam20C LOF and its effects on brain and/or CSF phosphoproteome in Raine Syndrome, which the authors incorrectly abbreviate as RS (see OMIM, Raine syndrome, RNS; 259775).
The manuscript attempts to ‘mine’ available databases or previously published manuscripts, and then describes in detail the biological functions of many of these proteins but does not provide any (independent) confirmatory evidence. The in-silico analyses thus suffer from a problem common to such studies, i.e., correlation but not causation/elicitation. The figures are generally of poor quality and/or resolution (especially Figs. 1-3) and do not adequately describe how the data sets were ‘mined’. The rationale for the separate Tables 2-4 is unclear, and many of the tables are missing information critical for proper interpretation (for example, see Table 6), thereby rendering it difficult to discern what the indicated numbers mean, the source data, and the relevant statistical analysis, etc.
The authors do not discuss that most cases of RNS (affected individuals) survive only days or weeks, and that missense, rather than amorphic (LOF), mutations are associated with non-lethal cases, i.e., those few individuals that survive to early childhood. There is no attempt to reconcile the nature of these missense mutations, their location(s) relative to known functional domains in Fam20C, predictions of their effects on Fam20C-kinase activity, regulation, locale, etc., and the possibility that survivors may in fact be the outcome of hypomorphic (partial LOF) mutations whose effects on the brain and/or CNS phospho-proteome will be difficult to predict a priori. This scenario is further compounded by the possibility that sub-lethal mutations in RNS may be linked to varying levels of compromised Fam20C kinase activity/regulation along with heterogeneity and/or biased biallelic expression (see compound heterozygosity noted in Simpson et al, 2009, Clin. Genet. 75: 271-276). Consequently, the correlative analysis lacks the robustness needed to make this case. The manuscript also suffers due to lack of a mechanistic link between Fam20C interactors, previously identified in other studies, as contributing factors to RNS, and incorrectly identifies databases such as MIM instead of OMIM. Lastly, the manuscript could benefit from copy editing. The manuscript is difficult to consider acceptable in its current state. Significantly revising the analytical rigor and copy editing appear to be warranted.
Author Response
Response to reviewer 1
Dear Reviewer
We attend to the observations you made to the manuscript, entitled “Potential role of protein kinase FAM20C on the brain in Raine syndrome, an in silico analysis”. Your observations were very helpful and enriched the manuscript, we were very happy correcting and editing it.
Moreover, the manuscript has been reviewed by an English-speaking native and all the manuscript abbreviations were reviewed. Below are the punctual responses to every observation you made. We hope it is complete, but we’ll be waiting if there is something else to correct.
All the authors want to thank you.
Best regards,
Carmen Palacios Reyes.
Comments and Suggestions for Authors
This manuscript attempts to correlate the pathophysiology of Raine Syndrome with sub-lethal mutations in Fam20C, expression of FAM20C targets and interactors in different human brain structures, and their roles in biological processes. As stated by the authors, the precise molecular defects downstream of biallelic Fam20C mutations in Raine Syndrome is not well understood but hypothesize that loss of Fam20C-kinase activity is expected to result in the hypo-phosphorylation of specific targets, alteration of the brain and/or CSF phosphoproteome(s), and thereby contribute to the developmental pathophysiology of this syndrome.
Overall Summary:
While the overall hypothesis is clearly stated and well justified, the analyses presented in the manuscript falls short of providing a significant mechanistic link between Fam20C LOF and its effects on brain and/or CSF phosphoproteome in Raine Syndrome, which the authors incorrectly abbreviate as RS (see OMIM, Raine syndrome, RNS; 259775).
Response:
RS was changed to RNS.
We consider that we are presenting potential FAM20C targets and interactors, as well as their potential significance in neural functions. Although we are not showing a specific and significant mechanism involved in neural pathogenesis, we did identify proteins with neural functions that participate in different pathways and could be involved in potential neural processes. We agree that experimental approaches are needed, such as functional assays to demonstrate those specific mechanisms involved in RNS neurological defects. The aim of this work is to give a general overview of potential and wide molecular processes, which could be involved in RNS neural pathogenesis. However, a deeper and functional analysis will be considered when resources are available.
The manuscript attempts to ‘mine’ available databases or previously published manuscripts, and then describes in detail the biological functions of many of these proteins but does not provide any (independent) confirmatory evidence. The in-silico analyses thus suffer from a problem common to such studies, i.e., correlation but not causation/elicitation. The figures are generally of poor quality and/or resolution (especially Figs. 1-3) and do not adequately describe how the data sets were ‘mined’. The rationale for the separate Tables 2-4 is unclear, and many of the tables are missing information critical for proper interpretation (for example, see Table 6), thereby rendering it difficult to discern what the indicated numbers mean, the source data, and the relevant statistical analysis, etc.
Response:
We identified through different available databases, proteins that interact with FAM20C. We intend to predict the interactor’s potential activity on neural tissues according to their brain expression and functions. Indeed, it does not prove or is evidence of the biological functions, but, we are only making predictions which could promote further functional studies confirming biological signaling or pathways involved.
We agree that in-silico analyses associate or correlate variables, and they are useless for cause identification.
We improved the figure quality and/or resolution and changed the description on how the data sets were ‘mined’.
Figure 1 was improved, adding a heatmap zoom, representing the genes with the highest and lowest ten percent expression.
We changed table 6. Now it is table 2, which now correlates with the text.
Additional information was added for tables 2-5.
The authors do not discuss that most cases of RNS (affected individuals) survive only days or weeks, and that missense, rather than amorphic (LOF), mutations are associated with non-lethal cases, i.e., those few individuals that survive to early childhood. There is no attempt to reconcile the nature of these missense mutations, their location(s) relative to known functional domains in Fam20C, predictions of their effects on Fam20C-kinase activity, regulation, locale, etc., and the possibility that survivors may in fact be the outcome of hypomorphic (partial LOF) mutations whose effects on the brain and/or CNS phospho-proteome will be difficult to predict a priori. This scenario is further compounded by the possibility that sub-lethal mutations in RNS may be linked to varying levels of compromised Fam20C kinase activity/regulation along with heterogeneity and/or biased biallelic expression (see compound heterozygosity noted in Simpson et al, 2009, Clin. Genet. 75: 271-276). Consequently, the correlative analysis lacks the robustness needed to make this case. The manuscript also suffers due to lack of a mechanistic link between Fam20C interactors, previously identified in other studies, as contributing factors to RNS, and incorrectly identifies databases such as MIM instead of OMIM. Lastly, the manuscript could benefit from copy editing. The manuscript is difficult to consider acceptable in its current state. Significantly revising the analytical rigor and copy editing appear to be warranted.
Response
We added the variant types of the patients listed in table 1. We identified the type of variants in lethal and non-lethal cases, including their location in the kinase domain and considered the amorphic and hypomorphic cases.
The compound heterozygous case reported by Simpson et al, 2009, in Clin. Genet. 75: 271-276) was considered. We identified the variants of the patient, they correspond to one inherited from the father and the other was a de novo variant, located in exons 2 and 3 respectively. They are not located in the kinase domain, as such, they can be hypomorphic variants producing FAM20C partial loss of function.
MIM database was corrected to OMIM.
Copy editing was done.

Reviewer 2 Report
The article by Palma-Iara et al. focuses on highlighting the potential role of FAM20C within the brain in the context of Raine syndrome by in silico analysis of its interactome. They summarize the current literature on Raine syndrome and attempt to identify the range of symptoms and defects of the patients. They also bring to focus several targets and interactors of FAM20C and describe their functions within the brain and their potential affect within the pathology of Raine syndrome.
While the work presented here is of potential interest for the field, I believe the manuscript needs to go through a major revision.
Multiple statements made in the introduction and the discussion sections are missing references or the ones provided are not relevant or sufficient. For example, in the introduction after they state the phosphorylation capacity of FAM20C for the S-x-Q-x-x-D-E-E motif without giving any relevant citations for it. The current reference to the work of Brunati et al., 2000 is not relevant or sufficient for the statement.
Other minor issues with the text from introduction include:
1. Line 112-114: The statement of 75% of phosphoproteins of the cerebrospinal fluid containing a S-x-E/pS motif is not backed by any citations.
2. Line 107: It would be very helpful to introduce the small integrin-binding ligand N-linked glycoprotein (SIBLING) family for better readability.
The results section needs to be improved in data readability and summarizing. In its current form there are several issues that affect text comprehension, such as:
1. Table 1 readability should be improved. In its current form, the labels of the header aren’t legible and the noted differences in the defects observed in patients aren’t distinguishable.
2. Figure 1 is not readable, and my suggestion would be to at least label differentially the top 10 genes they refer to as being of interest.
3. Table 3 and 4 are never referred to in the text and it doesn’t bear any description of the content.
4. Figure 2 has very little readability and it needs to be enlarged and of better quality.
5. At “2.2 List of FAM20C targets and interactors” in the results they quote a report from Tagliabracci of identified target proteins without any citation. Furthermore, the findings presented in this section are not mentioned to be summarized in any figure or table in the manuscript.
6. In the 2.6 section where they talk about identifying the top 20 genes, they present Table 6 with 26 highlights, and don’t specify the criteria for what is considered a top hit. For example TIMP1 has a value of 332 in Cerebral cortex but it isn’t considered a top hit, meanwhile HSPD1 has a 314.1 and is highlighted.
The discussion section would benefit from breaking it down into multiple subchapters, as in its current form the text is not easy to follow. I suggest dividing it into 4 sections comprised of: a summary for the structural neurological defects, one for functional defects, followed by a section for target interactors broken further for the two terms of interest that came up in the analysis and the genes of interest.
In the discussion section from the very beginning there are discrepancies between the findings they present in the table and what they mention in the text. Such is the case of the number of cases with cortex defects reported in the literature which they state it as 5 (line 270), afterwards as 4 (line 294), meanwhile in Table 1 they present a total of 6.
Other minor issues with the discussion text include:
1. Throughout the text there are different annotations of the MIM entries.
2. Line 431 should be supplemented with more published information on the causes for seizures in Raine syndrome.
3. Line 482 has AP3B1 mentioned twice.
4. Line 561 there is an odd citation of Jorissen et al., 2010 (ref. 83).
5. Line 582 a bad format of the citation of Hussain et al., 2019 (ref. 93)
The text throughout the manuscript needs language revision because there are multiple mistakes that make the text hard to comprehend. One such example is at line 404 that reads: “However, since the identification of genes and their roles in the brain have increased, it will continue.”
Author Response
Response to reviewer 2
Dear Reviewer
We attend to the observations you made to the manuscript, entitled “Potential role of protein kinase FAM20C on the brain in Raine syndrome, an in silico analysis”. Your observations were very helpful and enriched the manuscript, we were very happy correcting and editing it.
Moreover, the manuscript has been reviewed by an English-speaking native and all the manuscript abbreviations were reviewed. Below are the punctual responses to every observation you made. We hope it is complete, but we’ll be waiting if there is something else to correct.
All the authors want to thank you.
Best regards,
Carmen Palacios Reyes.
Comments and Suggestions for Authors
The article by Palma-Iara et al. focuses on highlighting the potential role of FAM20C within the brain in the context of Raine syndrome by in silico analysis of its interactome. They summarize the current literature on Raine syndrome and attempt to identify the range of symptoms and defects of the patients. They also bring to focus several targets and interactors of FAM20C and describe their functions within the brain and their potential affect within the pathology of Raine syndrome.
While the work presented here is of potential interest for the field, I believe the manuscript needs to go through a major revision.
Multiple statements made in the introduction and the discussion sections are missing references or the ones provided are not relevant or sufficient. For example, in the introduction after they state the phosphorylation capacity of FAM20C for the S-x-Q-x-x-D-E-E motif without giving any relevant citations for it. The current reference to the work of Brunati et al., 2000 is not relevant or sufficient for the statement.
Response:
Citations were reviewed and modified.
Other minor issues with the text from introduction include:
- Line 112-114: The statement of 75% of phosphoproteins of the cerebrospinal fluid containing a S-x-E/pS motif is not backed by any citations.
Response:
References were reviewed and modified.
- Line 107: It would be very helpful to introduce the small integrin-binding ligand N-linked glycoprotein (SIBLING) family for better readability.
Response:
The text was changed, and mor information of SIBLIBG proteins were added. The text was modified to: “SIBLING family members are the main FAM20C targets, which consists of osteopontin, bone sialoprotein, dentin matrix protein 1, dentin sialophosphoprotein and matrix extracellular phosphoglycoprotein, with actions in bone and dentin for proper mineralization.”
The results section needs to be improved in data readability and summarizing. In its current form there are several issues that affect text comprehension, such as:
- Table 1 readability should be improved. In its current form, the labels of the header aren’t legible and the noted differences in the defects observed in patients aren’t distinguishable.
Response:
The table was changed.
- Figure 1 is not readable, and my suggestion would be to at least label differentially the top 10 genes they refer to as being of interest.
Response:
The figure was changed. A zoom of the gene with highest expression was done, we improve the resolution of the image and new plots were added.
- Table 3 and 4 are never referred to in the text and it doesn’t bear any description of the content.
Response:
The tables are referred in the text now.
- Figure 2 has very little readability and it needs to be enlarged and of better quality.
Response:
The figure quality and resolution were improved.
- At “2.2 List of FAM20C targets and interactors” in the results they quote a report from Tagliabracci of identified target proteins without any citation. Furthermore, the findings presented in this section are not mentioned to be summarized in any figure or table in the manuscript.
Response:
The citation was added. A table was done, with the list of FAM20C target-interactors from Biogrid and Tagliabracci and classic targets. It was added as a supplement table.
- In the 2.6 section where they talk about identifying the top 20 genes, they present Table 6 with 26 highlights, and don’t specify the criteria for what is considered a top hit. For example TIMP1 has a value of 332 in Cerebral cortex but it isn’t considered a top hit, meanwhile HSPD1 has a 314.1 and is highlighted.
Response:
The test was modified, to show the top hit with statistical significance. Later, the level of expression is shown for cortex, white matter and cerebellum, were different values of expression can be observed.
The discussion section would benefit from breaking it down into multiple subchapters, as in its current form the text is not easy to follow. I suggest dividing it into 4 sections comprised of: a summary for the structural neurological defects, one for functional defects, followed by a section for target interactors broken further for the two terms of interest that came up in the analysis and the genes of interest.
Response:
We added sections as suggested. The following sections were described:
3.1 Structural and functional neurological defects in RNS
3.1.1 Structural brain defects
3.1.2 Functional brain defects
3.2 Brain expression of FAM20C target-interactors
In the discussion section from the very beginning there are discrepancies between the findings they present in the table and what they mention in the text. Such is the case of the number of cases with cortex defects reported in the literature which they state it as 5 (line 270), afterwards as 4 (line 294), meanwhile in Table 1 they present a total of 6.
Response
The frequency of defects was corrected in table 1 and in the text.
Other minor issues with the discussion text include:
- Throughout the text there are different annotations of the MIM entries.
Response
The MIM term was changed to OMIM.
- Line 431 should be supplemented with more published information on the causes for seizures in Raine syndrome.
Response:
Since few is described of seizures in patients with RNS, few data were added. The text changed to:
“Seizures in RNS cases can be related to low calcium levels in some, can be of different types including either focal or generalized kinds, but a specific cause is not described or identified.”
- Line 482 has AP3B1 mentioned twice.
Response:
It was corrected.
- Line 561 there is an odd citation of Jorissen et al., 2010 (ref. 83).
Response
It was corrected.
- Line 582 a bad format of the citation of Hussain et al., 2019 (ref. 93)
Response:
It was corrected.
The text throughout the manuscript needs language revision because there are multiple mistakes that make the text hard to comprehend. One such example is at line 404 that reads: “However, since the identification of genes and their roles in the brain have increased, it will continue.”
Response:
All the manuscript was language reviewed.
Copy editing was done.

Reviewer 3 Report
The authors described the structural and functional defects in RS; identified FAM20C targets and interactors, and analyzed molecular process, function and components. The following are my comments and critique:
1. The authors are suggested to examine the expression of FAM20C in brain tissues.
2. The authors are suggested to explain the significance of the different colors in table 1 to help readers better understand the results.
3. What results in the different expression levels in brain structures? The authors need to discuss it in depth.
4. The authors are suggested to list the top 10 FAM20C targets in the table.
5. The authors are suggested to select several significant FAM20C targets and explore their functions in cellular experiments.
Author Response
Response to reviewer 3
Dear Reviewer
We attend to the observations you made to the manuscript, entitled “Potential role of protein kinase FAM20C on the brain in Raine syndrome, an in silico analysis”. Your observations were very helpful and enriched the manuscript, we were very happy correcting and editing it.
Moreover, the manuscript has been reviewed by an English-speaking native and all the manuscript abbreviations were reviewed. Below are the punctual responses to every observation you made. We hope it is complete, but we’ll be waiting if there is something else to correct.
All the authors want to thank you.
Best regards,
Carmen Palacios Reyes.
Comments and Suggestions for Authors
The authors described the structural and functional defects in RS; identified FAM20C targets and interactors, and analyzed molecular process, function and components. The following are my comments and critique:
- The authors are suggested to examine the expression of FAM20C in brain tissues.
Response
It would be ideal to examine the expression of FAM20C in brain tissues during fetal development and adulthood. To date, it is not possible for us to do that, due to lack of resources. Therefore, we considered doing an in silico analysis for this manuscript, and we add data (from rnaseq database) of expression in different brain call types. We will consider it for future projects.
- The authors are suggested to explain the significance of the different colors in table 1 to help readers better understand the results.
Response
We added the explanation of the different color codes in table 1.
- What results in the different expression levels in brain structures? The authors need to discuss it in depth.
Response
We modified the text in order to discuss these data.
- The authors are suggested to list the top 10 FAM20C targets in the table.
Response
We modified table 6 and changed it to table 2, which now shows FAM20C targets with the highest brain expression (top 10%).
- The authors are suggested to select several significant FAM20C targets and explore their functions in cellular experiments.
Response
To date, we are not able to do that. This is the reason why this manuscript is only an in silico analysis. We would like to examine the expression and phosphorylation state of FAM20C targets such as CLU, ATP1A2, ATP1A1, PLD3, CALR among others, due to its relevance and association to disease, as well as those related to calcium brain homeostasis. We will consider it for future projects.

Round 2
Reviewer 1 Report
Potential role of protein kinase FAM20C on the brain in Raine syndrome, an in silico analysis. By Icela Palma-lara et al.
Revised manuscript:
This manuscript attempts to correlate the pathophysiology of Raine Syndrome with sub-lethal mutations in Fam20C, expression of FAM20C targets and interactors in different human brain structures, and their roles in biological processes. As stated by the authors, the precise molecular defects downstream of biallelic Fam20C mutations in Raine Syndrome is not well understood but hypothesize that loss of Fam20C-kinase activity is expected to result in the hypo-phosphorylation of specific targets, alteration of the brain and/or CSF phosphoproteome(s), and thereby contribute to the developmental pathophysiology of this syndrome.
Summary: The overall hypothesis is clearly stated and well justified, the analyses presented in the manuscript provides new links between Fam20C LOF and its effects on brain and/or CSF phosphoproteome in Raine Syndrome. Although the analyses do not uncover functional and/or mechanistic links between Fam20C targets and the pathophysiology of RNS, the revisions do provide a 'blueprint' for future investigations in RNS and are thus deemed suitable for publication in IJMS.
Author Response
Dear Reviewer
We attend to the observations you made to the manuscript, entitled “Potential role of protein kinase FAM20C on the brain in Raine syndrome, an in silico analysis”.
The manuscript has been reviewed by an English-speaking native. We’ll be waiting if there is something else to modify and made the modifications asked by each reviewer, and we are very pleased to improve the text which was enriched by their comments.
Thanks.
Best regards,
Carmen Palacios Reyes.

Reviewer 2 Report
The overall text readability was greatly improved by the language check and by reorganizing the discussions.
Tables are improved, and Figure 3 as well.
My reservations are regarding Figure 1 and 2 although improved in data presentation, still suffer from low quality/resolution. It is perhaps understandable the heat maps do not give much informative detail. But for the sake of the readership, authors should really consider enlarging data points and the text in at least Fig. 1D, and separating Fig. 2A from B and C and rotating 90° which would allow for proper detail.
Minor remarks are:
1. There’s a typo, the authors probably meant KDºKDº as they set the abbreviation in the text before.
Line 663:
“Only one non-lethal case with KºKº [69](reported 663 by Mameli et al 2020) was identified”
2. Some OMIM annotations are not fixed in the text. Such as:
Line 812:
“agenesis of corpus callosum, cardiac, ocular, and genital syndrome (618929), an entity featured by craniofacial”Line 847:“ADAM10 is associated to Alzheimer Disease 18 (615590), and Reticulate acropigmentation 847 of Kitamura (615537).”
Author Response
Dear Reviewer
We attend to the observations you made to the manuscript, entitled “Potential role of protein kinase FAM20C on the brain in Raine syndrome, an in silico analysis”. We are very pleased with the asked modifications because we consider the manuscript was improved and more understandable to the reader.
Also, the manuscript has been reviewed by an English-speaking native. Below are the punctual responses to every observation you made. We’ll be waiting if there is something else to modify.
Thanks.
Best regards,
Carmen Palacios Reyes.
Open Review
(x) I would not like to sign my review report
( ) I would like to sign my review report
Quality of English Language
( ) English very difficult to understand/incomprehensible
( ) Extensive editing of English language and style required
( ) Moderate English changes required
(x) English language and style are fine/minor spell check required
( ) I am not qualified to assess the quality of English in this paper
|
|
Can be improved |
Must be improved |
Not applicable |
|
|
Does the introduction provide sufficient background and include all relevant references? |
( ) |
( ) |
( ) |
(x) |
|
Are all the cited references relevant to the research? |
( ) |
( ) |
( ) |
(x) |
|
Is the research design appropriate? |
( ) |
( ) |
( ) |
(x) |
|
Are the methods adequately described? |
( ) |
( ) |
( ) |
(x) |
|
Are the results clearly presented? |
( ) |
( ) |
( ) |
(x) |
|
Are the conclusions supported by the results? |
( ) |
( ) |
( ) |
(x) |
Comments and Suggestions for Authors
The overall text readability was greatly improved by the language check and by reorganizing the discussions.
Tables are improved, and Figure 3 as well.
My reservations are regarding Figure 1 and 2 although improved in data presentation, still suffer from low quality/resolution. It is perhaps understandable the heat maps do not give much informative detail. But for the sake of the readership, authors should really consider enlarging data points and the text in at least Fig. 1D, and separating Fig. 2A from B and C and rotating 90° which would allow for proper detail.
Response
Figures were enlarged.
Figure 1 was separated in figure 1 and figure 2.
Figure 2 was separated in Figure 3 (molecular functions), figure 4 (molecular components) and figure 5 (molecular process).
Minor remarks are:
- There’s a typo, the authors probably meant KDºKDº as they set the abbreviation in the text before.
Line 663:
“Only one non-lethal case with KºKº [69](reported 663 by Mameli et al 2020) was identified”
Response
It has been corrected. Now the text reads:
“Only one non-lethal case with KDºKDº [69](reported 663 by Mameli et al 2020) was identified”
- Some OMIM annotations are not fixed in the text. Such as:
Line 812:
“agenesis of corpus callosum, cardiac, ocular, and genital syndrome (618929), an entity featured by craniofacial”
Response
It was corrected. Now the text is “(OMIM 618929)”.
Line 847:
“ADAM10 is associated to Alzheimer Disease 18 (615590), and Reticulate acropigmentation 847 of Kitamura (615537).”
Response
It was corrected to “ADAM10 is associated to Alzheimer Disease 18 (OMIM 615590), and Reticulate acropigmentation of Kitamura (OMIM 615537).”
Submission Date
23 February 2023
Date of this review
27 Apr 2023 13:29:42

Reviewer 3 Report
In the revised article, the authors modified the manuscript referred to the comments, and answered the questions comprehensively.
Author Response

(The authors gave the same response as above.)
